# Light-induced analgesia provides a drug-free optical method for pain relief via activation of TRAAK k$^+$ channels

Marion Bied[1,2,3], Arnaud Landra-Willm[1,2,3], Anne Amandine Chassot[1,2,3], Edward Francisco Mendez-Otalvaro[4], Benjamin Sueur[5], Kilian Roßmann[6], Elvira de la Peña [7], Pascal Fossat [5], Stephen J. Tucker [8,9], Jacques Noël[2,3,10], Wojciech Kopec [4,11], Felix Viana [7], Johannes Broichhagen [6], Eric Boué-Grabot [5] & Guillaume Sandoz [1,2,3] ✉

Pain management in animal experimentation is crucial for both ethical and scientific reasons, as unmanaged pain can distort physiological responses compromising data reliability. Current strategies are often invasive and pharmacology-based, introducing variability and confounding effects. Here, we present Light-Induced Analgesia, a drug-free, non-invasive method for pain relief in animals. We show that 365 nm illumination activates the pain-inhibitory TRAAK two-pore domain potassium (K2P) channel. This activation is driven by the oxidation of a native methionine at TRAAK's regulatory fenestration site, triggering a conformational switch from its inactive (down) to active (up) state. We further demonstrate that this mechanism can be transferred to other related K2Ps via a single-point mutation, rendering them light-sensitive. In rodents, gentle skin exposure to 365 nm is sufficient to activate endogenous TRAAK, silence nociceptors, and produce potent, long-lasting analgesia that outperforms standard treatments. Light-Induced Analgesia thus offers an effective, drug-free alternative that can enhance animal welfare and experimental reliability in preclinical research.

In preclinical rodent models, pain management is typically addressed using general or local analgesics, often in combination with anesthetics, and other techniques to minimize invasiveness[1]. Common drugs like opioids, non-steroidal anti-inflammatory drugs (NSAIDs), and anesthetics help reduce pain but may interfere with physiological and behavioral responses, potentially affecting the outcome of studies on immune function, inflammation, cognition, and/or metabolism[2–4]. These traditional approaches can introduce variability and confounding factors, impacting the reproducibility and interpretation of results. Despite these challenges, effective pain management remains essential for both ethical reasons and scientific accuracy. The development of non-invasive, drug-free pain relief methods is therefore critical, and in alignment with the refinement guideline of the 3Rs principle (replacement, reduction, refinement) that provides a framework to ensure more ethical practices in animal experimentation worldwide[5].

[1]Université Côte d'Azur, CNRS, INSERM, iBV, Nice, France. [2]Laboratories of Excellence, Ion Channel Science and Therapeutics, Nice, France. [3]Fédération Hospitalo-Universitaire InovPain, Cote d'Azur University, University Hospital Centre Nice, Nice, France. [4]Computational Biomolecular Dynamics Group, Max Planck Institute for Multidisciplinary Sciences, Göttingen, Germany. [5]Université de Bordeaux, CNRS, Institut des Maladies Neurodégénératives, UMR 5293, F-33000, Bordeaux, France. [6]Leibniz-Forschungsinstitut für Molekulare Pharmakologie (FMP), Berlin, Germany. [7]Instituto de Neurociencias de Alicante, Universidad Miguel Hernández-CSIC, San Juan de Alicante, Spain. [8]Department of Physics, Clarendon Laboratory, University of Oxford, Oxford, UK. [9]Kavli Institute for Nanoscience Discovery, University of Oxford, Oxford, UK. [10]Université Côte d'Azur, CNRS, INSERM, IPMC, Valbonne, France. [11]Department of Chemistry, Queen Mary University of London, London, UK. ✉e-mail: sandoz@unice.fr

Nociceptors play a pivotal role in pain circuits since they detect harmful or deleterious situations in the periphery and relay noxious information to the central nervous system, initiating the pain sensation[6]. The ideal method to induce analgesia would therefore specifically regulate their activity by controlling endogenous pain-related targets without the need for drugs, genetic manipulation, surgery, or other forms of intervention. In addition, this method would enable long-term experiments, be amenable to use in different species commonly found in laboratories, and be non-invasive.

In this work, we introduce Light-Induced Analgesia (LIA), an alternative method to alleviate pain in rodents in a spatiotemporally defined manner without the use of drugs. Through simple 365 nm illumination of the skin, LIA induces the oxidation of a native methionine residue of TRAAK, a pain-related Two-Pore-Domain Potassium (K2P) channel[7,8], leading to its strong activation. The resulting hyperpolarization of nociceptor free nerve endings silenced their activity and produced relevant levels of analgesia in freely moving rodents without requiring pharmacology, transfection, infection, or surgery.

## Results

### Choice of the stimulus and protein target to control pain

Since nociceptors are the primary entry points for pain signals and their free nerve endings are accessible to external stimuli, inhibition of their activity without pharmacological intervention constitutes a strategy for drug-free pain relief. This requires the identification of both a specific target protein - such as an ion channel - that controls nociceptor excitability, as well as a non-invasive stimulus capable of modulating the activity of this target.

Among proteins controlling the excitability of nociceptors, channels from the Two-Pore-Domain potassium channel (K2P) family support leak K$^+$ currents and serve as a brake for nociceptors' electrical excitability when they are active[9]. Dorsal Root Ganglion (DRG) and Trigeminal (TG) sensory neurons including nociceptors express a variety of K2Ps, including TRESK, TRAAK, TREK2, TWIK2, TREK1, THIK2, TASK1, TASK2, THIK1, and TASK3 in descending order of transcript abundance[10], as well as TWIK1[11]. Among those, only TRESK, TREK1, TREK2, TRAAK, TASK1, and TASK3 have been reported to be involved in (mechanical or thermal) noxious sensing[7,12-17], suggesting their functional role in the free nerve endings of DRG sensory neurons and making them promising targets for analgesia.

K2Ps, while being commonly considered as voltage-independent channels, are subjected to a complex multimodal regulation[18,19]. Indeed, regulatory factors of K2P channels range from physiological mediators (such as bioactive lipids, secondary messengers) to physicochemical parameters, including intra- or extracellular pH, mechanical pressure, and temperature, depending on the K2P subtype. However, light sensitivity of the pain-related K2P channels, particularly in the ultraviolet A (UV-A) range, has been overlooked in their initial characterization. Yet, UV-A light (320–400 nm), the least energetic class of light within the UV spectrum, can induce, mostly indirectly, chemical modification of biomolecules, thereby influencing their activity. Furthermore, UV-A light is able to penetrate the epidermis[20,21], which contains free nerve endings of nociceptors[22,23]. UV-A therefore represents a good potential stimulus to modify the activity of K2Ps localized in nociceptor nerve endings, influencing, in turn, the sensation of pain.

### Light-induced analgesia (LIA), a drug-free method to locally induce analgesia in rodents through activation of TRAAK currents by 365 nm illumination

To investigate the UV-A sensitivity of K2Ps expressed in DRG sensory neurons, whole-cell currents of HEK293T cells expressing a rodent (mouse) ortholog of each channel were recorded upon a 365 nm-centered illumination. Intriguingly, we observed that whereas no or

moderate current increases were observed for TREK1, TREK2, TASK1, TASK3 and TRESK, cells expressing TRAAK showed a robust ~8-fold current increase at 0 mV after 100 s of 365 nm illumination at 0.47 mW/mm$^2$ (Fig. 1a–c), at both room temperature (24 °C) and skin temperature (32 °C)[24] (Supplementary Fig. 1).

Given the reported expression of TRAAK in free nerve endings in the skin[7,8,25] and the penetration of UV-A in the epidermis[20,21], we reasoned that if a part of the skin is illuminated with 365 nm light, the activation of TRAAK channels would hyperpolarise the free nerve endings and decrease nociceptor excitability within the illuminated region, leading to local analgesia. We therefore tested this hypothesis on the paws of freely moving mice.

Due to light attenuation through the skin, only approximately 10 % of the energy from 365 nm light applied at the skin surface reaches the basal layer of the epidermis[20,21]. We therefore estimated that a 10 min 365 nm illumination of the skin with a power density of 1.39 mW/mm$^2$ (power of the lamp used in the animal facility) would deliver ~80 mJ/mm$^2$ of energy within the epidermis. This dose corresponds to one to two times the amount required to induce an 8-fold increase in TRAAK currents in HEK293T cells (Fig. 1a–c). Importantly, this illumination protocol, owing to its relatively short duration and low intensity, is harmless to mice, as confirmed by both macroscopic examination and histological analysis of 365 nm illuminated skin, which showed no signs of inflammation or tissue damage (Supplementary Fig. 2). Potential long-term side-effects (premature skin aging, DNA damage) due to the repeated application of this illumination protocol cannot however be fully excluded. Nevertheless, they are less of an issue in the scope of preclinical studies given the relatively short lifetime of rodents used in research settings.

Paws were therefore illuminated with 365 nm light for 10 min at 1.39 mW/mm$^2$, and their mechanical sensitivity was monitored using von Frey filaments. We found that our illumination protocol more than doubled the von Frey mechanical withdrawal threshold (Fig. 1d, f). This sensitivity decrease lasted for at least 1 h and disappeared within 3 h (Fig. 1d, f), thus demonstrating a significant analgesic effect of the 365 nm illumination of the skin.

To confirm the role of TRAAK in this 365 nm illumination-induced analgesic effect, we injected a solution containing a TRAAK channel inhibitor, RU-TRAAK2[26], in mice paws following 365 nm illumination of the skin (Fig. 1e). As expected, RU-TRAAK2 almost completely reversed the analgesic effect of the 365 nm illumination (Fig. 1e, g). Conversely, injection of a saline solution did not modify the mechanical withdrawal threshold increase obtained after illumination.

We therefore demonstrate that 365 nm illumination induces a current increase of the pain-related TRAAK channel and establish that 365 nm skin illumination constitutes a simple, drug-free and non-invasive approach for inducing localized analgesia, a method which we have named Light-Induced Analgesia (LIA). To explore the mechanisms underlying LIA more thoroughly, we next characterized the light sensitivity of TRAAK channels and its molecular determinants.

### 365 nm illumination irreversibly activates TRAAK channels in a time- and intensity-dependent manner

To fully characterize the light sensitivity of TRAAK channels, we first assessed its sensitivity throughout the electromagnetic spectrum. We observed that light-induced activation of TRAAK is specific to UV-A-shifted wavelengths, as blue (470 nm), green (505 nm) or yellow (565 nm) illuminations failed to increase TRAAK currents as much as UV-A light (365 or 385 nm) did (Fig. 2a, b). As shown in Fig. 1b, exposure at 365 nm triggers a progressive increase in TRAAK currents, which is linear with time over the first 100 s (for a light intensity of 0.47 mW/mm$^2$). Prolonged exposure showed nevertheless that TRAAK 365 nm current increase tends to reach > 15-fold increase after ~5 min of continuous illumination (Fig. 2c, Supplementary Fig. 3a). Beyond exposure time, we also investigated the effect of light intensity on channel

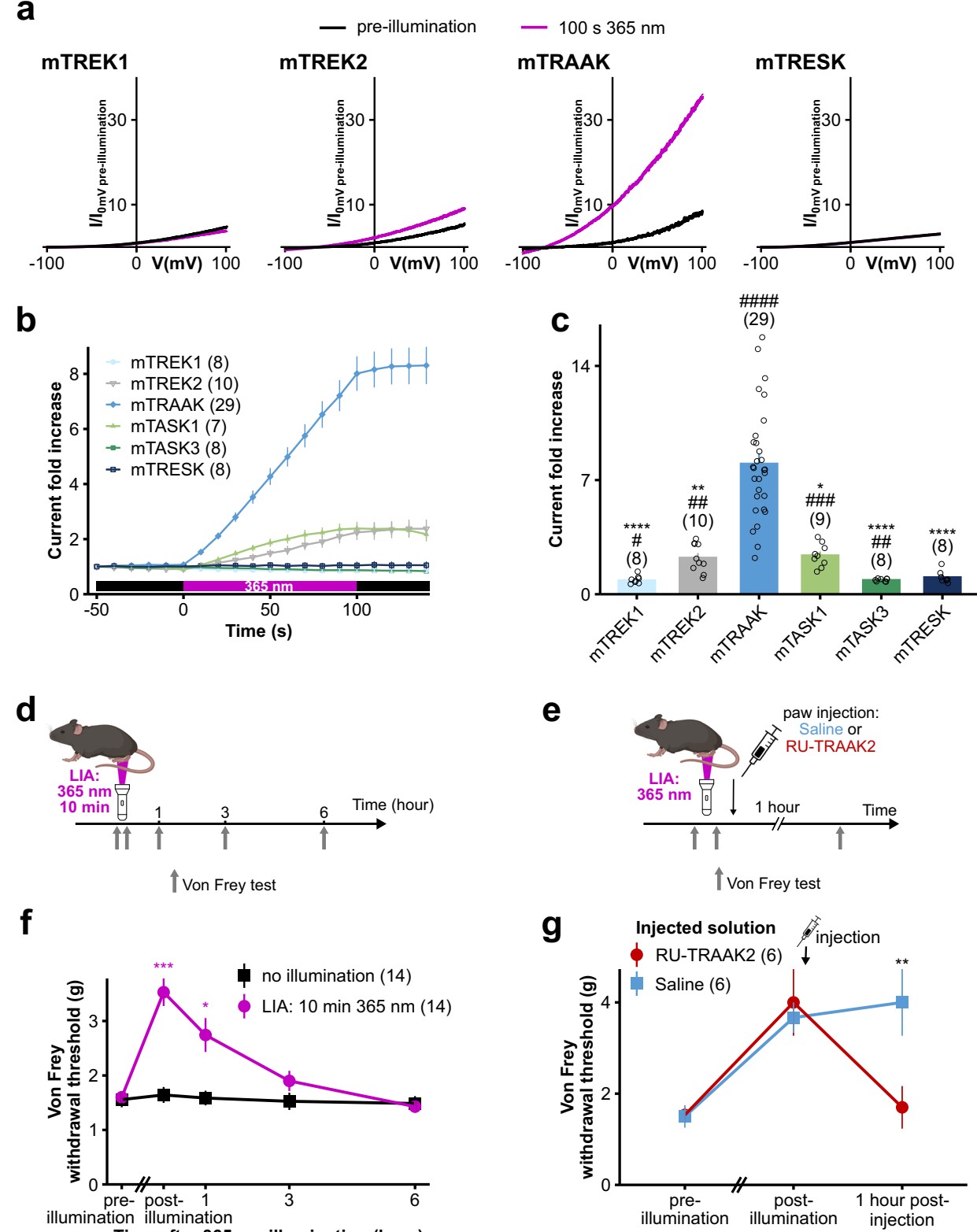

activation and found that TRAAK currents increase linearly with light intensity, at least within the tested range (Fig. 2d, e).

Once 365 nm illumination stops, TRAAK currents stabilize (Supplementary Fig. 3b) and remain elevated for at least 5 h. Currents start to reduce significantly after 8.5 h and return to baseline level after ~19 h at most (Fig. 2a). This strongly suggests

that light activation of TRAAK is irreversible at the channel level and that the delayed current decrease results from channel turnover in the plasma membrane. In support of this, we found that the timescale of TRAAK channel degradation matched that of TRAAK current decrease following 365 nm illumination (Supplementary Fig. 3c-e).

**Fig. 1 | Skin 365 nm illumination induces analgesia via activation of mTRAAK.**
**a** Representative normalized whole-cell current traces of HEK293T cells expressing mTREK1, mTREK2, mTRAAK, or mTRESK before and after a 100 s 365 nm-centered illumination. Currents were elicited by voltage-clamp ramps (−100 to +100 mV, 800 ms) and were normalized by the 0 mV current of the cell recorded before illumination. **b** Time course of channel responses to a 100 s 365 nm-centered illumination. Current fold increases were computed at 0 mV relative to the pre-illumination 0 mV current of the cell. **c** Current fold increases at 0 mV for each tested K2P channel after 100 s of 365 nm-centered illumination. Two independent statistical analyses were carried out: two-sided one-sample $t$ tests or two-sided Wilcoxon's signed rank tests comparing fold increases to 1 ($^{\#}p < 0.05$, $^{\#\#}p < 0.01$, $^{\#\#\#}p < 0.001$, $^{\#\#\#\#}p < 0.0001$ – mTREK1: $p = 0.04186$, mTREK2: $p = 0.00147$, mTRAAK: $p = 7.79 \times 10^{-12}$, mTASK1: $p = 0.00037$, mTASK3: $p = 0.00407$, mTRESK: $p = 0.74219$) and Kruskal–Wallis tests followed by Dunn's two-sided multiple comparison test comparing responses to that of mTRAAK (*$p < 0.05$, **$p < 0.01$, ***$p < 0.001$, ****$p < 0.0001$ – mTREK1: $p = 1.59 \times 10^{-7}$, mTREK2: $p = 0.00814$, mTASK1: $p = 0.02771$, mTASK3: $p = 2.64 \times 10^{-7}$, mTRESK: $p = 2.54 \times 10^{-6}$).

**d**, **e** Diagrams explaining the experimental timelines of (**f**) and (**g**), respectively (created in BioRender). **f** Time course of the analgesic effect of a 10 min 365 nm illumination (LIA) of wild-type mice paws quantified through von Frey withdrawal threshold. For each paw series independently, statistical analyses compare the von Frey withdrawal threshold at each time point to that of the pre-illumination time point using a Friedman test followed by Dunn's two-sided multiple comparison test (*$p < 0.05$, **$p < 0.01$, ***$p < 0.001$ – for LIA treated paws: post-illumination: $p = 0.0001$, 1 h: $p = 0.0136$, 3 h: $p = 0.5407$, 6 h: $p > 0.9999$, for non-illuminated paws: Friedman test – $p = 0.5325$). **g** Reversion of 365 nm skin illumination-induced analgesia (LIA) by the injection of the RU-TRAAK2 TRAAK inhibitor. Statistical analysis consists of a two-way Anova test followed by a Holm–Sidak's multiple comparison test, comparing von Frey withdrawal thresholds of saline or RU-TRAAK2 injected paws for each time point (*$p < 0.05$, **$p < 0.01$ – pre-LIA: $p = 0.9625$, post-LIA: $p = 0.8695$, 1 h post-injection: $p = 0.008$). Data shown are mean ± s.e.m. The number of cells or paws used in each experiment is indicated on the figure. Source data are provided in the Source Data file.

## Light sensitivity of TRAAK is species-dependent

Since all mammalian subgroups express a highly conserved TRAAK channel ortholog (*KCNK4* in humans), we wondered if its light sensitivity is universal among different mammalian species (Fig. 2h). (Of note, all previous results were obtained on the *Mus musculus* TRAAK channel.) Interestingly, we found this property conserved in other rodents (rats—*Rattus norvegicus*), as well as in species from groups phylogenetically distant from mice, such as chiroptera (Jamaican fruit bats—*Artibeus jamaicensis*) and marsupials (koalas—*Phascolarctos cinereus*). However, TRAAK light sensitivity is not universal as TRAAK channels from primates (humans—*Homo sapiens*), carnivorans (cats—*Felis catus*), artiodactyls (cows—*Bos taurus*), and lagomorphs (rabbits—*Oryctolagus cuniculus*) failed to exhibit a strong current increase upon 365 nm illumination (Fig. 2h, i, Supplementary Fig. 4).

We therefore highlight that TRAAK currents irreversibly increase in response to UV-A shifted illumination in a time-, and intensity-dependent manner. However, while this characteristic is shared by diverse mammalian subgroups, it is not universal.

## 365 nm illumination activates TRAAK channels independently of cytoplasmic processes

We then aimed at determining the mechanism responsible for TRAAK light activation and first tested the potential involvement of cytoplasmic processes. To that end, we assessed the effect of 365 nm illumination on TRAAK currents recorded from inside-out patches, a configuration in which the patch is excised from the cytoplasm. As shown in Fig. 2f, g, a 30 s-lasting 365 nm illumination increases TRAAK inside-out currents by 5-fold, while the current of non-illuminated patches remained stable. These results indicate that 365 nm illumination induces TRAAK channel activation, independently of any cytoplasmic factors or other signaling processes. In particular, these results rule out the possibility of an increase in the number of channels at the membrane in response to 365 nm illumination.

## A methionine residue in TM2 (M134) is necessary and sufficient for the light activation of TRAAK

We next aimed at identifying the region of TRAAK responsible for its 365 nm activation. Based upon the clear difference in light sensitivity between mouse TRAAK (mTRAAK) and human TRAAK (hTRAAK) channels (Fig. 2i), we carried out a mTRAAK-hTRAAK chimeric approach.

As shown in Fig. 3a, c (and Supplementary Fig. 5a), chimeric channels containing the first half of mTRAAK were sensitive to 365 nm illumination, as is mTRAAK. Conversely, channels containing the second half of mTRAAK were insensitive to 365 nm light, similarly to hTRAAK. The light-sensitive part of mTRAAK, therefore, lies within the first half of the channel.

Similar to other K2Ps, the first half of TRAAK channels contains an extensive extracellular domain between the end of the first transmembrane segment and the beginning of the selectivity filter, termed the extracellular cap[27] and suggested as important in K2P regulation[28]. We therefore built chimeric human–mouse TRAAK channels in which the extracellular cap along with the P1 loop is swapped to assess the involvement of the extracellular cap in mTRAAK light sensitivity. As shown in Fig. 3b, c (and Supplementary Fig. 5b), mTRAAK channel with hTRAAK cap and P1 loop was strongly sensitive to 365 nm light similarly to mTRAAK, while hTRAAK with mTRAAK cap and P1 loop remained insensitive to 365 nm illumination. Therefore, the light-sensitive part of mTRAAK lies outside of the extracellular cap and P1 loop, either within the beginning of the channel up to the extracellular cap or within the second transmembrane segment (TM2).

Due to the high degree of sequence conservation, comparison of sequences of the two putative light-sensitive domains of TRAAK from two UV-A-sensitive (mouse and rat) and two UV-A-insensitive species (humans and cow) highlighted a single amino acid difference at position 134 (in mouse TRAAK) that may be critical for light sensitivity (Fig. 3d). We therefore tested the involvement of methionine 134 (M134) in mTRAAK light activation. As shown in Fig. 3e–g, the M134I mutation in mTRAAK completely abolished its sensitivity to 365 nm light, while the I133M mutation in hTRAAK (I133 in hTRAAK is homologous to mTRAAK M134) was sufficient to confer 365 nm sensitivity on hTRAAK. These results clearly demonstrate that this single residue (M134) is not only necessary, but also sufficient, for TRAAK UV-A activation.

TRAAK is also very closely related to two other K2P channels, TREK1 and TREK2, neither of which are UV-A sensitive (Fig. 1a–c), nor have a methionine at this position (Fig. 3h). Interestingly, we found that introducing a methionine at this position in TREK2 (F197M) could also endow light sensitivity to TREK2 (Fig. 3i–k), thus confirming the role for a methionine at the 134 homologous position in K2P channel UVA-activation.

Following the identification of the unique amino acid responsible for TRAAK light sensitivity (M134), we next examined the effect of 365 nm UV-A illumination on M134 and its role in channel activation.

## TRAAK light activation relies on M134 oxidation

Since methionine possesses a sulfur atom that can be oxidized, thereby transforming into methionine sulfoxide which features an additional double bond oxygen (Supplementary Fig. 6a), we hypothesized that the light activation of TRAAK results from the oxidation of M134 by reactive oxygen species (ROS) resulting from UV-A illumination.

To test the M134 oxidation hypothesis, we assessed whether the generation of ROS, independently of UV-A illumination, could activate mTRAAK. For this purpose, we engineered the CA-ATTO-MB2

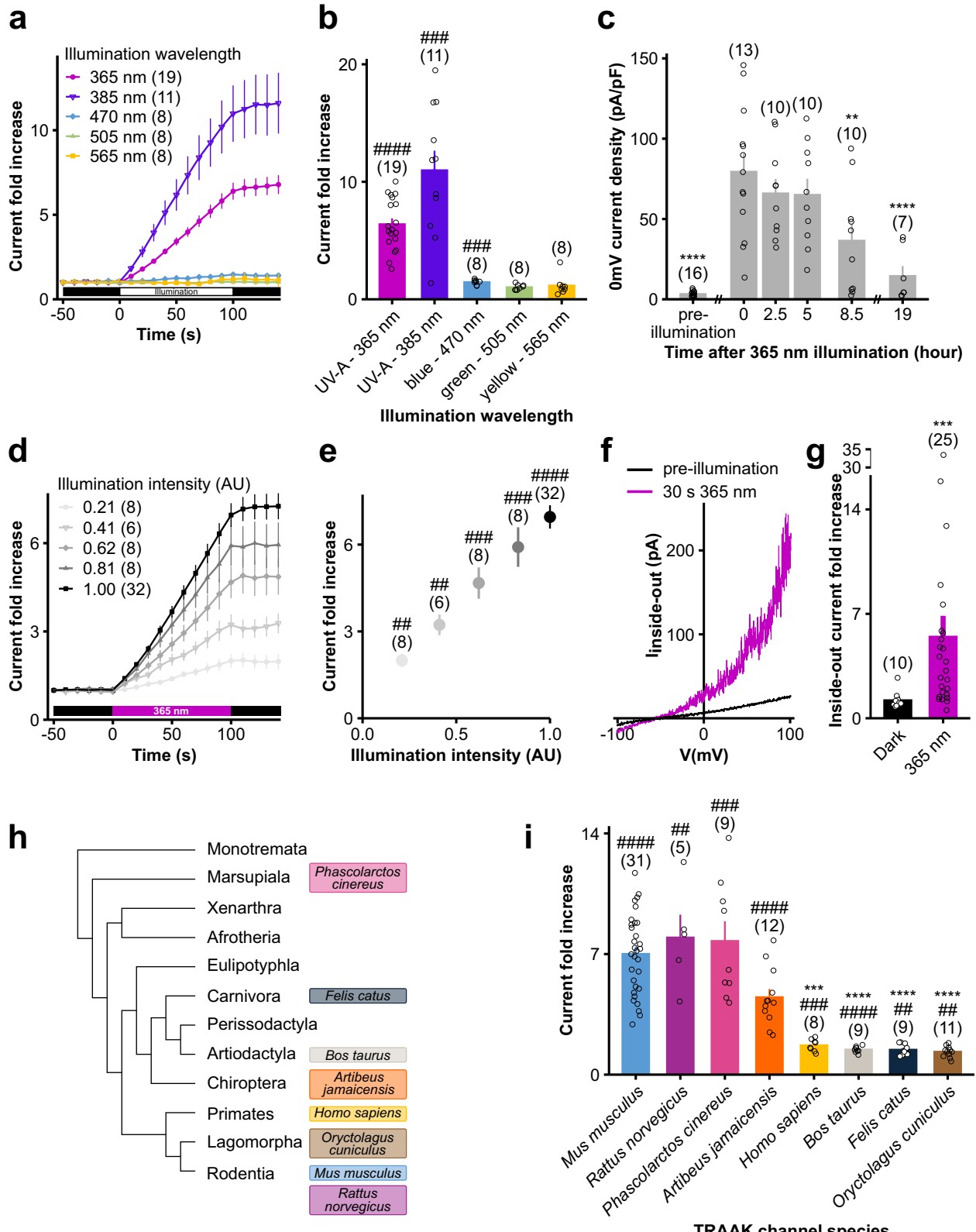

compound. It contains the ATTO-MB2 reagent, a derivative of methylene blue dye, that generates singlet oxygens when illuminated with long-wavelength light (550-700 nm)[29,30]. Through its binding to Halo-tagged proteins with its added chloroalkane group (CA), the design of the CA-ATTO-MB2 compound therefore allows both spatial and temporal control of singlet oxygen production in response to light (Fig. 4a). As shown in Fig. 4b–d, after incubation with CA-ATTO-

MB2, the current of cells expressing Halo-tagged mTRAAK channels increased strongly upon yellow (565 nm) light illumination, while staying stable in the dark. This effect was not observed in Halo-tagged mTRAAK M134I channels (the UV-A-insensitive mTRAAK mutant) or without prior incubation with CA-ATTO-MB2, thus demonstrating that generation of singlet oxygens in the vicinity of mTRAAK is sufficient to activate the channel, likely through M134 oxidation.

**Fig. 2 | 365 nm illumination irreversibly activates TRAAK channels in a time-, intensity- and species-dependent manner, independently of cytoplasmic processes. a** Time course of mTRAAK responses to different wavelength illuminations. Current fold increases were computed at 0 mV relative to the pre-illumination 0 mV current of the cell. **b** mTRAAK current fold increases at 0 mV after 100 s illuminations at different wavelengths. Statistical analyses compare the fold increases to 1 using two-sided one sample t-tests or two-sided Wilcoxon's signed rank tests ([#]$p < 0.05$, [##]$p < 0.01$, [###]$p < 0.001$, [####]$p < 0.0001$ – 365 nm: $p = 2.86\ 10^{-9}$, 385 nm: $p = 0.000133$, 470 nm: $p = 0.00044$, 505 nm: $p = 0.73932$, 565 nm: $p = 0.84375$). **c** mTRAAK current recovery to baseline after a 10 min 365 nm illumination at 1.39 mW/mm². Statistical analyses compare currents to the 0 h time point using Welch's one-way ANOVA followed by Dunnett's test (*$p < 0.05$, **$p < 0.01$, ***$p < 0.001$, ****$p < 0.0001$ – pre-illumination: $p = 2.29 \times 10^{-9}$, 2.5 h: $p = 0.70044$, 5 h: $p = 0.64800$, 8.5 h: $p = 0.00286$, 19 h: $p = 3.23 \times 10^{-5}$). **d** Time course of mTRAAK current increase induced by 365 nm-centered illuminations at different intensities. Current fold increases were computed at 0 mV relative to the pre-illumination 0 mV current of the cell. **e** Relationship between mTRAAK current fold increase at 0 mV and 365 nm illumination intensity. Here, illuminations lasted 100 s. Statistical analyses compare fold increases to 1 using two-sided one sample t-tests or a two-sided Wilcoxon's signed rank test ([#]$p < 0.05$, [##]$p < 0.01$, [###]$p < 0.001$, [####]$p < 0.0001$ – 0.21 AU: $p = 0.00343$, 0.41 AU: $p = 0.00154$, 0.62 AU: $p = 0.00026$, 0.83 AU: $p = 0.00018$, 1 AU: $p = 1.38 \times 10^{-15}$). **f** Representative traces of currents recorded in an inside-out macropatch excised from a mTRAAK expressing HEK293T cell before and after a 30 s 365 nm illumination. Inside-out currents were elicited by a voltage-clamp ramp (−100 to +100 mV, 800 ms). **g** Inside-out current fold increases at 40 mV after 30 s of recording in the dark or upon 365 nm illumination. Statistical analyses compare the two recording conditions using a two-sided Mann–Whitney test (***$p = 0.00012$). **h** Phylogenetic tree of mammals, showing the position of the species whose TRAAK channels are studied in (**i**). **i** Current fold increases at 0 mV of TRAAK mammalian orthologs after a 100 s 365 nm-centered illumination. Two independent statistical analyses were carried out: two-sided one-sample t-tests or two-sided Wilcoxon's signed rank tests comparing fold increases to 1 ([#]$p < 0.05$, [##]$p < 0.01$, [###]$p < 0.001$, [####]$p < 0.0001$ – Mus musculus: $p = 4.16 \times 10^{-15}$, Rattus norvegicus: $p = 0.00622$, Phascolarctos cinereus: $p = 0.00035$, Artibeus jamaicensis: $p = 1.85 \times 10^{-5}$, Homo sapiens: $p = 0.00070$, Bos taurus: $p = 4.60 \times 10^{-5}$, Felis catus: $p = 0.00374$, Oryctolagus cuniculus: $p = 0.00655$) and Kruskal–Wallis test followed by Dunn's two-sided multiple comparison test comparing responses to that of mTRAAK (*$p < 0.05$, **$p < 0.01$, ***$p < 0.001$, ****$p < 0.0001$ – Rattus norvegicus: $p > 0.99999$, Phascolarctos cinereus: $p > 0.99999$, Artibeus jamaicensis: $p = 0.43828$, Homo sapiens: $p = 0.00055$, Bos taurus: $p = 9.35 \times 10^{-6}$, Felis catus: $p = 4.85 \times 10^{-6}$, Oryctolagus cuniculus: $p = 1.11 \times 10^{-7}$). Data shown are mean ± s.e.m. The number of recorded cells or patches is indicated in the figure. Source data are provided in the Source Data file.

Finally, since the oxidative reaction of methionine into methionine sulfoxide increases the electronic environment in its vicinity (Supplementary Fig. 6a), we hypothesized that the effect of M134 oxidation into methionine sulfoxide can be mimicked by mutating M134 by either partially or totally charged residues. As expected, the basal currents of mTRAAK M134E (fully charged residue) and mTRAAK M134Q (polar residue) are 45 and 39-fold higher than that of wild-type mTRAAK, respectively (Fig. 4e, Supplementary Fig. 6b, c). Similar to mTRAAK M134E or M134Q mutants, we found that introducing a positively charged residue (M134R) activates mTRAAK but not as much as a partially or fully negatively charged residue (Fig. 4e, Fig. 6b, c). Our results thus confirm that introduction of a charged or polar environment at the position of M134 is associated with an activated state of mTRAAK, no matter the nature of the charge.

## UV-A-induced M134 oxidation switches TRAAK to its conductive "up" conformation

M134 is located within the TM2 helix of mTRAAK, at a site known to be critical for the regulation of TREK and TRAAK channels. Indeed, M134 lines the interface between the TM2 and TM4 helices that form the fenestration. The latter corresponds to a hydrophobic pocket open to the surrounding membrane and creates a passageway between the pore and the lipid bilayer (Fig. 4f). Movement of the TM4 helix from the "down" to "up" conformation causes this fenestration to disappear and switches TREK and TRAAK channels into a more active state[31–33]. Mutations at the interface between TM2 and TM4 have also been shown to regulate channel activity[34,35], highlighting that a subtle chemical change within this region can significantly impact channel activity.

We therefore wondered whether M134 oxidation modifies the equilibrium between the up and down conformations of TRAAK to activate the channel and used molecular dynamics simulations and free energy calculations to assess the energetics of the up-to-down state transition of the hTRAAK I133M mutant (I133 in hTRAAK is homologous to M134 in mTRAAK) (Fig. 3d). We found that oxidation of M133 increased the relative free energy difference of the up-to-down transition by 6.6 ± 2.8 kJ/mol compared to the non-oxidized channel, indicating that M134 oxidation stabilizes the more active "up" state of the channel (Fig. 4f, Supplementary Fig. 7).

These results therefore demonstrate that 365 nm light-activation of TRAAK relies on the oxidation of only one methionine, methionine 134, which triggers a conformational switch of TRAAK from the low-conductive "down" state to the high-conductive "up" state.

## 365 nm illumination of the skin decreases nociceptor excitability ex vivo and in vivo, constituting the basis of LIA

After determining the molecular mechanism underlying TRAAK activation by 365 nm light, we next investigated how activation of TRAAK by 365 nm illumination of the skin (LIA) leads to analgesia in rodents (Fig. 1d–g).

As mentioned previously, UV-A light (including 365 nm light) can penetrate the epidermis[20,21], which contains free nerve endings of nociceptors[22,23]. Since TRAAK has been reported to be expressed in nociceptors' free nerve endings of the skin[7], we hypothesized that illuminating a specific area of the skin in rodents would activate TRAAK channels of nerve endings located in that region. This activation would enhance the outward potassium (K⁺) current from the nerve endings, leading to their hyperpolarization. Such hyperpolarization would decrease their excitability and therefore decrease the sensitivity at the level of the illuminated region, i.e. create local analgesia.

We therefore characterized the effect of 365 nm illumination of the skin on the excitability of nociceptors within the illuminated region. We first assessed the effect of 365 nm illumination on nociceptor free nerve endings ex vivo, by comparing the excitability of C fibers before and after illumination of their receptive field in nerve-skin preparations[36] (Fig. 5a). After a 10-min 365 nm illumination, the electrical threshold required to activate C fibers increased significantly by more than 2-fold (Fig. 5b, c), demonstrating that UV-A illumination reduces nociceptor excitability.

We also investigated the effect of skin exposure to 365 nm light in vivo on the response of spinal dorsal horn (SDH) neurons following electrical stimulation of their receptive field[37] (Fig. 5d). SDH neurons receive inputs from C fibers within the spinal cord before relaying the information through the ascending pain pathway. As in the nerve-skin experiment, we found that the electrical stimulation threshold required to trigger a C fiber response strong enough to activate SDH neurons almost doubled following a 10-min 365 nm illumination of the corresponding receptive field (Fig. 5e, f). Furthermore, the evoked response, i.e. the number of delayed C-spikes evoked by 1, 2 or 3 mA stimulations was reduced by 2- to 3-fold following 365 nm illumination of the skin (Fig. 5g, h). 365 nm illumination of the skin therefore reduces the strength of nociceptive signals conveyed to the spinal

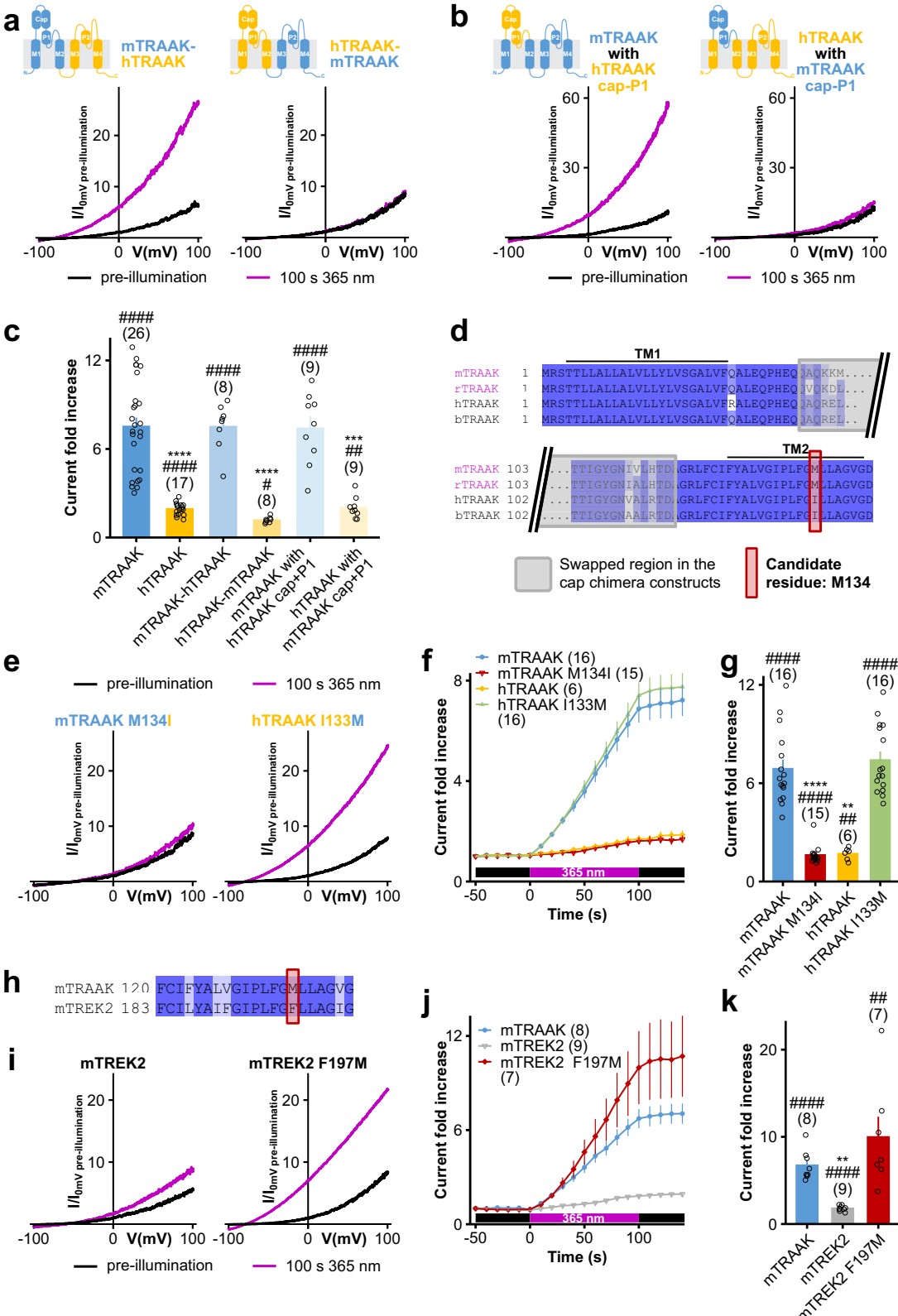

cord and the central nervous system, explaining the analgesic effect of skin 365 nm illumination at the level of the pain pathway.

We therefore demonstrate that LIA results from the activation of TRAAK channels (Fig. 1e, g), within nociceptor free nerve endings. The resulting increase of TRAAK currents is sufficient to significantly reduce nociceptor excitability, thereby inducing local analgesia.

## LIA is an efficient and drug-free method to locally induce analgesia in mice subjected to pain

To further validate LIA as an effective strategy for pain relief, we examined its analgesic effect in mice subjected to chronic pain using two models: the commonly used spared nerve injury (SNI) model of neuropathic pain[38] and the *Trek1−/−-Trek2−/−* double knock-out (KO) mice model which exhibit hyperexcitable nociceptors[16,39]. In the

**Fig. 3 | Methionine 134 is necessary and sufficient for light-activation of TRAAK.**
**a, b** Representative normalized whole cell current traces of HEK293T cells expressing human–mouse TRAAK chimeric channels before and after a 100 s 365 nm-centered illumination. The cartoons indicate channel composition. Currents were elicited by voltage-clamp ramps (−100 to +100 mV, 800 ms) and were normalized by the 0 mV current of the cell recorded before illumination. **c** Current fold increases at 0 mV of human–mouse TRAAK chimeric channels after a 100 s 365 nm-centered illumination. Refer to the cartoons in (**a**) and (**b**) for channel composition. Two independent statistical analyses were carried out: two-sided one-sample t-tests or two-sided Wilcoxon's signed rank tests comparing fold increases to 1 (#$p < 0.05$, ##$p < 0.01$, ###$p < 0.001$, ####$p < 0.0001$ – mTRAAK: $p = 7.49\ 10^{-11}$, hTRAAK: $p = 6.95 \times 10^{-8}$, mTRAAK-hTRAAK: $p = 9.42 \times 10^{-6}$, hTRAAK-mTRAAK: $p = 0.03404$, mTRAAK with hTRAAK cap+P1: $p = 4.67 \times 10^{-5}$, hTRAAK with mTRAAK cap+P1: $p = 0.00272$) and Kruskal–Wallis test followed by Dunn's two-sided multiple comparison test comparing responses to that of mTRAAK (*$p < 0.05$, **$p < 0.01$, ***$p < 0.001$, ****$p < 0.0001$ – hTRAAK: $p = 4.49 \times 10^{-6}$, mTRAAK-hTRAAK: $p > 0.99999$, hTRAAK-mTRAAK: $p = 1.68 \times 10^{-7}$, mTRAAK with hTRAAK cap+P1: $p > 0.99999$, hTRAAK with mTRAAK cap+P1: $p = 0.00030$). **d** Alignment of segments of TRAAK channels from Mus musculus (mTRAAK), Rattus norvegicus (rTRAAK) (both light-sensitive), Homo sapiens (hTRAAK) and Bos taurus (bTRAAK) (both light-insensitive)

potentially involved in UV-A activation, as inferred from the chimeric study. The position of methionine M134 is highlighted in red. (TM1, TM2: transmembrane regions 1 and 2). **e** Similar to (**a**) for mTRAAK M134I or hTRAAK I133M. **f** Time course of responses of mTRAAK, mTRAAK M134I, hTRAAK and hTRAAK I133M to a 100 s 365 nm-centered illumination. Current fold increases were computed at 0 mV relative to the pre-illumination 0 mV current of the cell. **g** Similar to (**c**) for mTRAAK, mTRAAK M134I, hTRAAK, hTRAAK I133M (statistical comparisons to 1 (represented with #): mTRAAK: $p = 2.40 \times 10^{-8}$, mTRAAK M134I: $p = 6.10 \times 10^{-5}$, hTRAAK: $p = 0.00597$, hTRAAK I133M: $p = 3.03 \times 10^{-9}$, statistical comparisons to mTRAAK (represented with *): mTRAAK M134I: $p = 7.57 \times 10^{-6}$, hTRAAK: $p = 0.00598$, hTRAAK I133M: $p > 0.99999$). **h** Alignment of mTRAAK and mTREK2 second transmembrane segments, with mTRAAK M134 highlighted in red. **i** Similar to (**a**) for mTREK2 and mTREK2 F197M. **j** Similar to f for mTRAAK, mTREK2 and mTREK2 F197M. **k** Similar to (**c**) for mTRAAK, mTREK2 and mTREK2 F197M (statistical comparisons to 1 (represented with #): mTRAAK: $p = 3.23 \times 10^{-5}$, mTREK2: $p = 5.36 \times 10^{-5}$, mTREK2 F197M: $p = 0.00838$, statistical comparisons to mTRAAK (represented with *): mTREK2: $p = 0.00351$, mTREK2 F197M: $p = 0.92843$). Data shown are mean ± s.e.m. The number of recorded cells is indicated in the figure. Source data are provided in the Source Data file.

context of neuropathic pain, 365 nm illumination of the paw skin doubled the von Frey mechanical threshold of SNI mice and therefore alleviated the mechanical allodynia induced by SNI (Fig. 6a, b). Likewise, for *Trek1⁻/⁻-Trek2⁻/⁻* double KO mice, the analgesic effect induced by the 365 nm illumination of the paw was even stronger than for wild-type mice (3-fold increase of the Von Frey withdrawal threshold; Fig. 6a, c) and lasted as long (Fig. 6d, h).

### LIA is a promising strategy to alleviate pain in mice, more effective than conventional analgesics
To position LIA among commonly used treatments for pain relief, we compared the efficacy of LIA to Emla cream and ibuprofen. While the lidocaine-prilocaine-based Emla cream is a widely used local analgesic applied at sites of injuries or surgical incisions, ibuprofen, which counts among non steroidal anti-inflammatory drugs (NSAIDs), is administered systemically to limit prostaglandin synthesis and therefore prevent inflammation[1,40]. In terms of intensity and duration, we found that the effect of LIA in naïve animals surpasses that of either ibuprofen or Emla cream (Fig. 6d, f, g). Similarly to naïve animals, the pain-relief effect of LIA exceeds the effects observed with both ibuprofen and Emla in the *Trek1⁻/⁻-Trek2⁻/⁻* double KO mice chronic pain model (Fig. 6d, h, i).

### LIA is effective in rats
To evaluate whether LIA could be extended to other species commonly used in preclinical research, we tested its efficacy in rats, a well-established model in laboratory experimentation. Notably, the rat version of TRAAK possesses a methionine at the position homologous to M134 (Fig. 3d) and displays light sensitivity (Fig. 2i). As in mice, 365 nm illumination induced a two-fold increase of the von Frey paw withdrawal threshold that persisted for at least 1 h (Fig. 6e). These results confirm that LIA is also effective in rats, thus extending further its applicability to another widely used laboratory animal model.

Overall, these results highlight the superior performance of LIA compared to conventional pharmacological approaches and demonstrate the robustness, reproducibility and efficacy of the analgesic effect induced by the 365 nm illumination of the skin in commonly used laboratory animals.

## Discussion
In this study, we report that the TRAAK K2P channel in certain species, in particular rat and mouse, is strongly activated by UV-A light. We demonstrate that this activation results from the oxidation of the methionine residue homologous to M134 in mTRAAK, positioned in

the TM2 transmembrane helix. This methionine is not only necessary, but also sufficient to confer light sensitivity on hTRAAK and on other related K2P members such as TREK2 which lack a methionine at this position. Molecular dynamics simulations reveal that oxidation of this methionine modifies a key regulatory interface between TM2 and TM4, favoring the more active "up" conformation of the channel. More importantly, we show that gentle skin 365 nm illumination is sufficient to activate TRAAK channels at the free nerve endings of nociceptors, silencing their activity, and producing local analgesia in freely moving naïve mice and rats as well as in rodent models of neuropathic pain (Fig. 7). This method aligns with the refinement aspect of the 3Rs principle (Replacement, Reduction, Refinement) in animal experimentation, offering a drug-free tool for analgesia in rodent pre-clinical studies.

In line with previous structural studies[27,41–43], our findings highlight the unique regulatory importance of the TM2/TM4 interface that forms the fenestration site within the TREK/TRAAK subfamily of K2P channels[27,32,44]. We demonstrate how subtle modification in this interface can regulate channel activity by impacting the energetics of the up/down conformation transitions of the channel. However, whether the UV-A sensitivity of certain TRAAK orthologs is physiologically relevant remains unresolved. Considering the high expression of TRAAK in the retina[45] and the nocturnal behavior of species whose TRAAK channels are UV-A sensitive (Fig. 2i), it is tempting to speculate that this property may confer an evolutionary advantage to nocturnal species, possibly acting on retinal activity in the daytime. Of note, K2P channel activity (namely TASK3) revealed to be important in retinal visual sensitivity[46].

In this study, we employed 365 nm wavelength to control native TRAAK channel activity with high spatiotemporal resolution given by the beam of light. Contrary to current light-control techniques that depend on genetics (optogenetics) or pharmacology (photopharmacology), this photophysiological approach allows the functional role of TRAAK and the physiological consequences of its activation to be explored in a non-invasive and non-pharmacological manner, without the need for genetically engineered channels[47–49]. Using this strategy, our study achieves spatiotemporal control of pain relief in freely moving animals, positioning TRAAK as a promising target for analgesic drug development.

From a broader perspective, our work paves the way for the development of a new generation of optogenetic tools enabling long-term silencing of neurons. In the field of optogenetics, the development of proteins to hyperpolarise neurons and inhibit their activity remotely with light has proved challenging, with the first endogenous

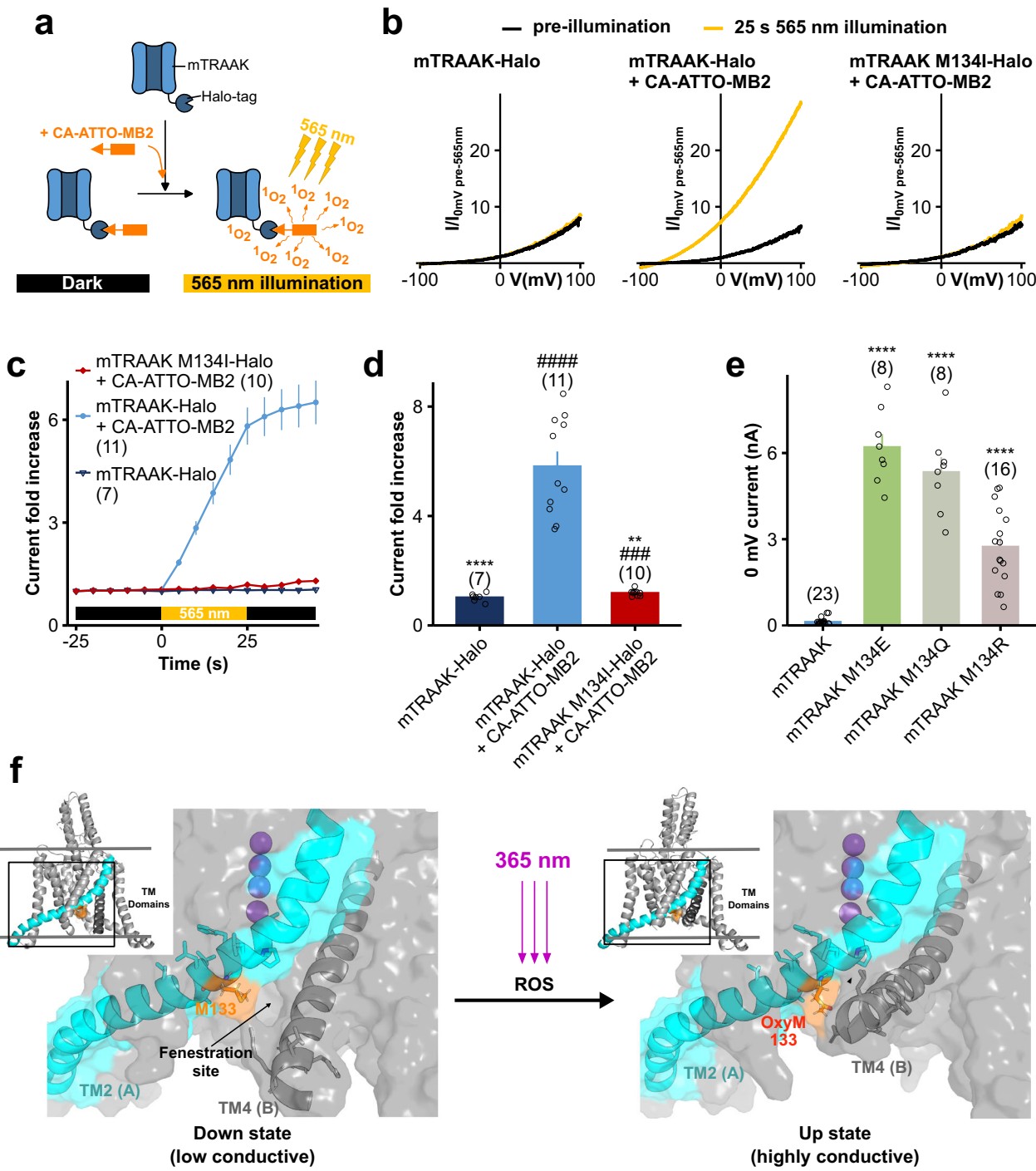

potassium channelrhodopsin discovered in 2022[50]. While tools like light-gated proton or anion pumps achieve short-term neuronal inhibition[51], the engineering of a potassium channel coupled to a photoreceptive LOV domain, BLINK2, allowed neuronal silencing for tens of minutes[52]. In contrast, based on our findings, we propose a strategy to engineer potassium channels that would be irreversibly activated by light, enabling the long-term (several hours) silencing of circuits upon a single 365 nm light pulse. Our strategy simply consists in using mTRAAK or introducing a methionine at the position homologous to M134 in a K2P channel with a low or silent basal activity. Such tools present therefore several advantages compared to the existing ones.

Finally, effective pain management in rodents is essential in animal experimentation to ensure scientific integrity and fulfill proper ethical and regulatory requirements intended to improve animal welfare. According to the IACUC (Institutional Animal Care and Use Committee), most methods available to induce analgesia in rodents are systemic and require the administration of analgesic drugs, that are either non-steroidal anti-inflammatory drugs (NSAIDs), like meloxicam and carprofen, or opioids such as buprenorphine or butorphanol. Only lidocaine and bupivacaine can be injected locally to relieve pain. However, all these techniques are invasive, as they require injection or administration of a drug and therefore introduce potential confounding effects, compromising the accuracy of scientific conclusions drawn from preclinical animal studies. In contrast, LIA bypasses these issues and appears as a non-invasive and drug-free alternative technique to provide effective and reproducible local analgesia in rodents without altering physiological or behavioral responses. LIA thereby

**Fig. 4 | Light activation of TRAAK relies on the oxidation of methionine 134 which switches TRAAK to its conductive "up" conformation. a** Diagram highlighting the properties of the CA-ATTO-MB2 compound, enabling the spatio-temporal control of singlet oxygen generation (created in BioRender). **b** Representative normalized whole-cell current traces of HEK293T cells expressing Halo-tagged mTRAAK or mTRAAK M134I before and after a 25 s 565 nm illumination, as cells were previously incubated or not during 30 min in 1 μM CA-ATTO-MB2. Currents were elicited by voltage-clamp ramps (−100 to +100 mV, 800 ms) and were normalized by the 0 mV current of the cell recorded before illumination. **c** Time course of responses of HEK293T cells expressing Halo-tagged mTRAAK or mTRAAK M134I to a 25 s 565 nm illumination, as the cells were previously incubated or not during 30 min in 1 μM CA-ATTO-MB2. **d** Current fold increases at 0 mV of Halo-tagged mTRAAK or mTRAAK M134I expressing HEK cells previously incubated or not in 1 μM CA-ATTO-MB2, after a 25 s 565 nm illumination. Two independent statistical analyses were carried out: two-sided one sample t-tests comparing 0 mV current fold increases to 1 ($^{\#}p < 0.05$, $^{\#\#}p < 0.01$, $^{\#\#\#}p < 0.001$, $^{\#\#\#\#}p < 0.0001$ – mTRAAK-Halo: $p = 0.66326$, mTRAAK-Halo + CA-ATTO-MB2: $p = 4.57 \times 10^{-6}$, mTRAAK M134I-Halo + CA-ATTO-MB2: $p = 0.00038$) and Kruskal–Wallis test

followed by Dunn's multiple comparison test comparing fold increases to that of Halo-mTRAAK expressing cells incubated in 1 μM CA-ATTO-MB2 (*$p < 0.05$, **$p < 0.01$, ***$p < 0.001$, ****$p < 0.0001$ – mTRAAK-Halo: $p = 3.16 \times 10^{-5}$, mTRAAK M134I-Halo + CA-ATTO-MB2: $p = 0.00205$). **e** Comparison of the basal 0 mV currents of mTRAAK, mTRAAK M134E, mTRAAK M134Q and mTRAAK M134R. Statistical analyses compare the 0 mV currents of mTRAAK mutants to that of wild-type mTRAAK using a Kruskal–Wallis test followed by Dunn's multiple comparison test (***$p < 0.001$, ****$p < 0.0001$ – mTRAAK M134E: $p = 6.87 \times 10^{-8}$, mTRAAK M134Q: $p = 1.90 \times 10^{-6}$, mTRAAK M134R: $p = 0.00031$). **f** Structural models of hTRAAK I133M with M133 in its reduced (left) or oxidized form (right) impacting the ("down" or "up") conformational state of the channel. Insets show the ribbon structure of hTRAAK I133M, with the black outline indicating the zoomed-in region of the TM2-TM4 interface. As highlighted in the figure, this interface undergoes substantial changes upon the down to up conformation switch of the channel. The transmembrane helix TM2(A) is highlighted in cyan, and TM4(B) is shown in dark gray. In the down state (left), the fenestration site is indicated with an arrow. Data shown are mean ± s.e.m. The number of recorded cells is indicated in the figure. Source data are provided in the Source Data file.

---

enhances scientific rigor and aligns with international ethical frameworks, particularly the refinement guideline of the 3Rs principle (Replacement, Reduction, Refinement)[5]. Moreover, since LIA requires only a 365 nm light illumination system, it is cost-effective and easily implementable in animal facilities.

We anticipate LIA will be employed in various contexts, ranging from the treatment of acute pain during procedures on unanesthetized animals to both minor or major surgeries. Indeed, in the context of surgeries, when animals are immobilized, LIA can be applied at the level of skin incisions easily with a simple portable 365 nm lamp. Similarly, for major surgeries, LIA may be applied preoperatively, intraoperatively, or postoperatively and even directly to internal organs during the procedure, by illuminating the exposed tissue. This approach could help reduce pain throughout the recovery phase. In these contexts, if drug administration does not interfere with experimental outcomes, LIA may even be used alongside conventional analgesics. In that case, since LIA is based on light and corresponds to a novel therapeutic modality, distinct from NSAIDs or opioids, we can anticipate that its effect would add up to that of drugs. LIA applications also concern routine procedures on freely moving animals, such as multiple injections, blood sampling, tissue collection for genotyping (ear, toe, or tail clipping), that are carried out today with little or no analgesia. In that case, LIA implementation requires the development of suits, handling tunnels or light plates equipped with 365 nm LEDs for light delivery at specific skin sites.

LIA therefore has clear potential to become an ethical and validated pain management tool that can improve both animal welfare and data reliability.

## Methods
### Molecular biology
Used mouse and human K2P clones are the following: mTRAAK: XP_006526782.1, mTREK1: AAC53005.2, mTREK2: NP_001303594.1, mTRESK: NP_997144.1, mTASK1: NP_034738.1, mTASK3: XP_006520831.1, hTRAAK: NP_001304019.1. The *Rattus norvegicus* TRAAK (XP_006230696.1) clone was purchased from genescript and the TRAAK clones from all other species (*Phascolarctos cinereus* TRAAK: XP_020823625.1, *Artibeus jamaicensis* TRAAK: XP_036996052.1, *Bos taurus* TRAAK: XP_024843418.1, *Felis catus* TRAAK: XP_019668181.1, *Oryctolagus cuniculus* TRAAK: XP_051689071.1) were synthesized by genecust. Chimeras and point mutations were built using standard PCR procedures (namely overlapping PCRs). Finally, to generate the Halo tagged versions of mTRAAK and mTRAAK M134I, the Halo tag was added in frame at the 3′ end of the corresponding DNA coding sequences by PCR. The primers used for the PCRs were synthetized by Sigma Aldrich (USA), and their sequences are provided in the Source Data file. All DNA

sequences of interest were inserted in the pIRES2EGFP plasmid for transfection and confirmed with DNA sequencing (microsynth).

### Cell culture and transfection
HEK (human embryonic kidney) 293 T cells were purchased from ATCC (CRL-11268) and grown in high glucose DMEM (Dulbecco's Modified Eagle Medium) (Gibco) supplemented with 10% decomplemented fetal bovine serum, 100 U/mL penicillin and 100 U/mL streptomycin (1% Pen-Strep, Gibco) in humidified incubators with 5% $CO_2$ at 37 °C. Cells were passaged and plated twice per week on 35 mm diameter dishes and typically used from passages 10 to 40. Cells were transiently transfected with the aforementioned plasmids using the standard calcium phosphate transfection method (3.6 μg of DNA for 35 mm diameter dishes with cells at 75% confluency).

### Patch-clamp electrophysiology
**Recordings.** Electrophysiology experiments were carried out in the whole-cell and inside-out conventional configurations, 24 h after transfection at room temperature (22 ± 2 °C). Patch pipettes were pulled from borosilicate glass capillaries using a two-stage puller (PC-100, Narishige) and had a resistance ranging from 2.5 to 5 MΩ. For whole-cell patch-clamp recordings, pipettes were filled with the intracellular solution containing (in mM): 155 KCl, 5 EGTA, 3 $MgCl_2$, 10 HEPES, pH 7.3 with KOH, and cells were bathed in the extracellular solution consisting of (in mM): 150 NaCl, 5 KCl, 2 $CaCl_2$, 10 HEPES, pH 7.4 with NaOH. For inside-out patch-clamp recordings, pipettes contained the previously described extracellular solution. The gigaseal and patch excision were performed as cells were bathed in the extracellular solution, before perfusing the intracellular solution, enabling the recording of inside-out currents. Currents were elicited in the voltage-clamp mode with voltage-ramps (from −100 to 100 mV, 800 ms), followed by a 0 mV step lasting 100 ms and the cell (or patch) was maintained at −80 mV otherwise. Signals were amplified with a MultiClamp 700B amplifier (Axon Instruments, Molecular Devices Electrophysiology), low-pass filtered at 10 kHz and digitized with an Axon Digidata 1550B digitizer (Axon Instruments, Molecular Devices Electrophysiology) with a 20 kHz sampling rate. Recordings and analysis were performed using the pCLAMP 10.7 software.

**Illumination systems and protocols.** Apart from the reversibility experiment (Fig. 2c), the illumination of patched cells was ensured by mounted LEDs (365 nm: M365L3, 385 nm: M385L3, 470 nm: M470L3, 505 nm: M505L3, 565 nm: M565L3, Thorlabs) fixed on the microscope used for patch recordings. Therefore, the LED beam was focused on the patched cell through the inverted microscope objective, resulting

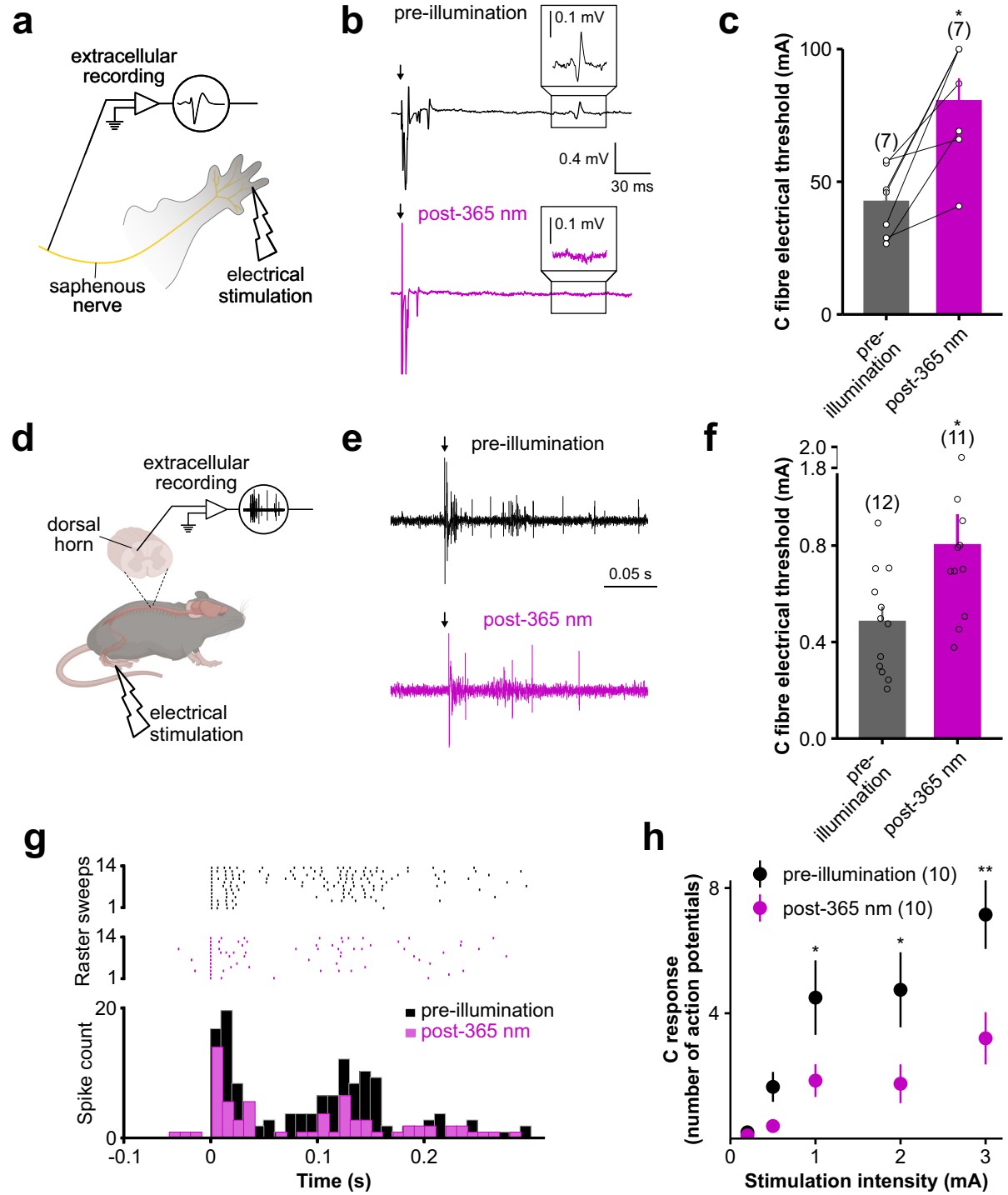

in a $0.47\,\text{mW/mm}^2$ intensity unless specified otherwise. For the experiment testing mTRAAK sensitivity to several wavelengths, the maximal $0.47\,\text{mW/mm}^2$ intensity recorded for the 365 nm LED was used as a reference to tune the power of the other 385, 470, 505 and 565 nm LEDs. Of note, because of the inverted configuration of the microscope, the light beam traverses the plastic dish containing the cells, reducing the effective light intensity applied to the cell. For standard experiments, 10 365 nm-light pulses lasting 10 s were applied to the patched cell maintained at −80 mV. 2 light pulses were separated by 1.5 s, allowing the recording of the current of the cell in the dark after each light pulse. For inside-out recordings, 5 s light pulses at 365 nm were applied.

For the reversibility experiment, whole plates of cells incubated in the extracellular solution (used for patch) were uniformly illuminated for 10 min using the external portable 365 nm lamp used for animal experiments (see the corresponding section). The lamp was placed 1 cm above the dish and set at its maximum intensity ($1.39\,\text{mW/mm}^2$). At the end of the illumination, cells were incubated for variable times in the dark at 37 °C, 5% $CO_2$, in standard culture medium.

For the CA-ATTO-MB2 experiment, the 565 nm LED set at its maximal power (intensity of $0.60\,\text{mW/mm}^2$ immediately after the objective) was used to generate the illumination beam. In this experiment, 565 nm light pulses lasted 5 s and were separated by 1.4 s of dark, allowing the recording of the cell's current.

**Fig. 5 | 365 nm illumination of the skin decreases the excitability of C-fibers.**
**a** Diagram of the nerve-skin preparation set-up used for extracellular recordings of
C-fibers within the saphenous nerve (created in BioRender). **b** Representative
nerve-skin recordings of C-fiber activity evoked by a 26.65 mA electrical stimulation
(arrow) before and after a 10 min 365 nm illumination of its receptive field. Insets
highlight the failure to record evoked C fiber activity following 365 nm illumination.
**c** Comparison of electrical stimulation thresholds of C-fibers before and after a
10 min 365 nm illumination of their receptive fields. Electrical thresholds before
and after illumination were compared using a two-sided Wilcoxon's signed rank
test (*$p = 0.0156$). **d** Diagram of in vivo extracellular recordings of spinal dorsal horn
(SDH) neuron activity evoked by the electrical stimulation of their receptive field
(created in BioRender). **e** Representative recordings of SDH neuron activity evoked
by a 4 mA electrical stimulation (arrow) before and after a 10 min 365 nm illumi-
nation of the corresponding receptive field. Note the decreased number of action
potentials following illumination. **f** Comparison of the electrical stimulation

threshold required to record activity in SDH neurons before and after a 10 min
365 nm illumination of their receptive field. Electrical thresholds before and after
illumination were compared using a two-sided Wilcoxon's signed rank test
(*$p = 0.0172$). **g** Representative raster plots (top) and superimposed histograms
(bottom) of SDH neuron activity elicited by a 4 mA electrical stimulation of the
corresponding receptive field, before (black) and after (purple) its illumination at
365 nm during 10 min. **h** C-fiber responses to different electrical stimulation
intensities before and after a 10 min 365 nm illumination of the paw, quantified by
the number of action potentials recorded in SDH neurons. Statistical analysis
consists in a two-way Anova test followed by a Holm–Sidak's multiple comparison
test, comparing C fiber response before and after illumination for each stimulation
intensity (*$p < 0.05$, **$p < 0.01$-0.2 mA: $p = 0.9255$, 1.65 mA: $p = 0.4285$, 4.5 mA:
$p = 0.0436$, 4.75 mA: $p = 0.0238$, 7.15 mA: $p = 0.0018$). Data shown are mean ± s.e.m.
The number of recorded neurons is indicated in the figure. Source data are pro-
vided in the Source Data file.

## CA-ATTO-MB2 experiment
**General experimental details on chemical synthesis and analysis.**
All chemical reagents and solvents for synthesis were purchased from
commercial suppliers (Sigma-Aldrich, Roth, ATTO-TEC) and were used
without further purification if not stated otherwise.

NMR spectra were recorded at 300 K in deuterated solvents on a
Bruker AV-III spectrometer using a room- temperature 5 mm broad-
band probe equipped with one-axis self-shielded gradients and cali-
brated to residual solvent peaks ($^1$H/$^{13}$C in ppm): MeOD-d$_4$ (3.31/49.00)
Multiplicities are abbreviated as follows: s = singlet, d = doublet, t =
triplet, q = quartet, p = pentet, h = heptet, br = broad, m = multiplet.
Coupling constants $J$ are reported in Hz. Spectra are reported based on
appearance, not on theoretical multiplicities derived from structural
information.

UPLC-UV/Vis for purity assessment was performed on an Agilent
1260 Infinity II LC System equipped with Agilent SB-C18 column
(1.8 μm, 2.1 × 50 mm). Buffer A: 0.1% FA in H$_2$O Buffer B: 0.1% FA acet-
onitrile. The typical gradient was from 10% B for 1.0 min → gradient to
95% B over 5 min → 95% B for 1.0 min with 0.45 mL/min flow or from
30% B for 1.0 min → gradient to 95% B over 5 min. Retention times ($t_R$)
are given in minutes (min).

Preparative or semi-preparative HPLC was performed on an Agi-
lent 1260 Infinity II LC System equipped with columns as followed:
preparative column –Reprospher 100 C18 columns (10 μm: 50 ×
30 mm at 20 mL/min flow rate; semi-preparative column – 5 μm: 250 ×
10 mm at 4 mL/min flow rate. Eluents A (0.1% TFA in H$_2$O) and B (0.1%
TFA in MeCN) were applied as a linear gradient. Peak detection was
performed at maximal absorbance wavelength.

For HRMS, samples were analyzed on Orbitrap Fusion mass
spectrometer (Thermo Fisher Scientific). MS scans were acquired in a
range of 350 to 1500 m/z. MS1 scans were acquired in the Orbitrap with
a mass resolution of 120,000 with an AGC target value of 4e5 and
50 ms injection time. MS2 scans were acquired in the ion trap with an
AGC target value of 1e4 and 35 ms injection time. Precursor ions with
charge states 2-4 were isolated with an isolation window of 1.6 m/z and
40 sec dynamic exclusion. Precursor ions were fragmented using
higher-energy collisional dissociation (HCD) with 30% normalized
collision energy.

## CA-ATTO-MB2 synthesis.
An Eppendorf tube was charged with ATTO
MB2-NHS ester (ATTO-TEC: #AD MB2-31) (400 μg, 915 nmol, 1.0 equiv.)
in 500 μL DMSO and DIPEA (0.64 μL, 3.66 μmol, 4.0 equiv.) and CA-NH$_2$
(306 μg, 1.37 μmol, 1.5 equiv.) was added subsequently. The mixture
was vortexed and allowed to incubate for 30 min before it was quen-
ched by addition of 2 μL of acetic acid and diluted with 1 mL dH$_2$O. RP-
HPLC (MeCN:H$_2$O + 0.1% TFA = 10:90 to 90:10 over 45 min) provided
178 μg of CA-ATTO MB2 (318 nmol, 35%) as a blue powder after lyo-
philization in 29% yield.

**HRMS** (ESI): calc. for C$_{29}$H$_{42}$ClN$_4$O$_3$S$^+$ [M]$^+$: 561.2661, found:
561.2671.

$^1$**H NMR** (600 MHz, MeOD-d$_4$): $\delta$ [ppm] = 7.99 (dd, $J = 9.7$, 1.2 Hz,
2H), 7.54 (d, $J = 9.5$ Hz, 1H), 7.52 (d, $J = 2.8$ Hz, 1H), 7.50 (d, 2.8 Hz, 1H),
7.42 (s,1H), 7.40 (d,$J = 2.8$ Hz, 1H), 3.76 (t,$J = 7.7$ Hz, 2H), 3.60 − 3.57 (m,
2H), 3.57 − 3.53 (m, 4H), 3.52 (t, $J = 6.7$ Hz, 2H), 3.44 (t, $J = 6.6$ Hz, 2H),
3.41 (s, 6H) 3.39 (t, $J = 5.4$ Hz, 2H), 3.38 (s, 3H), 2.35 (t, $J = 6.9$ Hz, 2H),
2.04 (p, $J = 7.31$, 2H), 1.72 (p, $J = 7.06$ Hz, 2H), 1.54 (p, $J = 7.07$ Hz, 2H),
1.42 (m, 2H), 1.37 − 1.35 (m, 2H), 5.61 (s, 2H), 4.62 (s, 2H).

See Supplementary Fig. 8 for the chemical structure of CA-ATTO-
MB2 (formal name: *N*-(7-((4-((2-(2-((6-Chlorohexyl)oxy)ethoxy)ethyl)
amino)−4-oxobutyl)(methyl)amino)−3*H*-phenothiazin-3-ylidene)-*N*-
methylmethanaminium 2,2,2-trifluoroacetate).

CA-ATTO-MB2 incubation on cells:

CA-ATTO-MB2 was dissolved in dimethyl sulfoxide (DMSO) at
1 mM, and diluted in DMEM medium to 1 μM. Cells were incubated in
the CA-ATTO-MB2 solution in the dark at 37 °C, 5% CO2 for 30 min,
prior to electrophysiological recordings.

## Animal experiments
**Mice and rat strains.** All animal experiments were conducted according
to national and international guidelines and have been approved by the
local ethical committees (Ministère de la Recherche, de l'Enseignement,
et de l'Innovation CIEPAL - APAFIS #33381-2021093008408503 v8,
APAFIS #21135-2019061914043519 v3, Ethics Committee from Uni-
versidad Miguel Hernández). The C57BL/6 J wild-type or *Trek1-Trek2*
double KO mouse and Wistar rat breeders and offspring were maintained
on a 12 h light/dark cycle with constant temperature (21–23 °C), humidity
(45–50%), and food and water *ad libitum* at the animal facilities of Institut
de Biologie Valrose, of Bordeaux Neurocampus – Université de Bor-
deaux, or of Miguel Hernández University of Elche. Behavioral experi-
ments were performed on 7 to 13-week old male or female mice weighing
20–30 g and on adult Wistar male and female rats weighing 346 ± 14 g.

Knock-out mice lacking *Trek1* and *Trek2* were generated as
described previously[53]. Null mutations were backcrossed against the
C57BL/6 J inbred strain for more than 10 generations prior to estab-
lishing the breeding cages to generate subjects for this study.

**Single C-fiber recordings.** The isolated skin-saphenous nerve pre-
paration and single-fiber recording technique were performed as
described previously[36]. Briefly, the skin of the right hind paw along with
the saphenous nerve of mice was carefully dissected and kept in the
recording chamber under laminar perfusion of warm (-30 °C) synthetic
interstitial fluid (SIF) containing (in mM): 120 NaCl, 3.5 KCl, 2 CaCl$_2$, 0.7
MgSO$_4$, 5 NaHCO$_3$, 1.7 NaH$_2$PO$_4$, 9.5 NaGluconate, 7.5 sucrose, 10 Hepes,
5.5 glucose, pH = 7.4 with NaOH. The skin was placed "outside-down",
exposing the interior of the skin to the experimenter. The nerve was
threaded through a hole into a separate recording chamber. Fine fila-

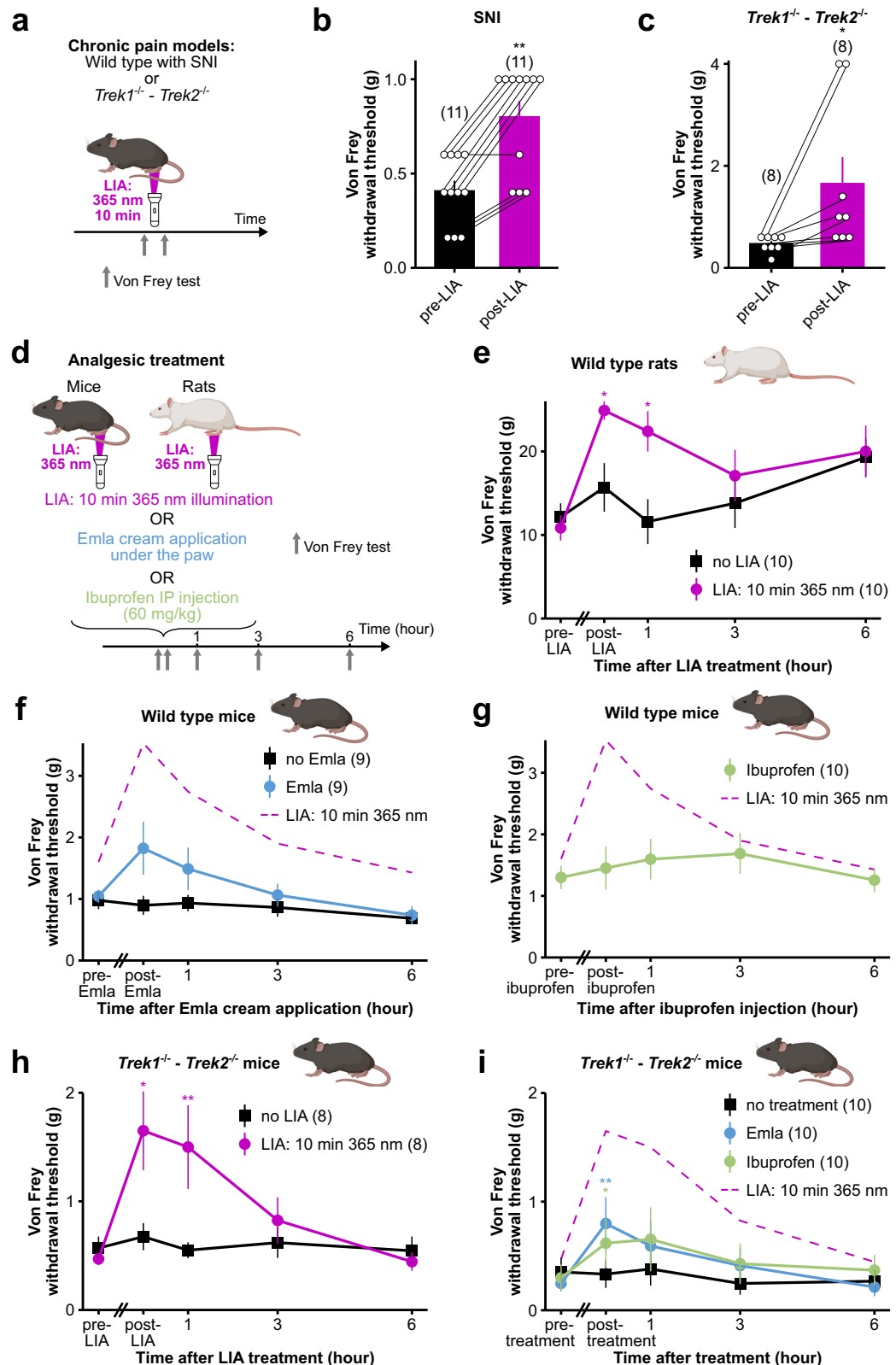

ments were then isolated from the nerve and placed on a gold recording electrode connected to a DAM-80 AC differential amplifier (WPI) to record one or several C fiber responses. Using a glass rode, the corresponding receptive field was identified and isolated with a thick-wall ring. An electrical stimulation electrode was placed within the ring to determine the electrical stimulation threshold of the recorded C fibers, as well as the conduction velocity of the fiber (fibers with a conduction velocity of less than 1.3 m/s were included in the study). The receptive field was then continuously illuminated for 10 min with the portable 365 nm lamp used in animal experiments (refer to the corresponding section), and the electrical stimulation threshold was assessed once again after the end of the illumination for comparison. All recordings were digitalized with the Digidata 1322 A acquisition system (Axon Instrument), using the pClamp 10 software.

**Fig. 6 | LIA is a robust method for pain relief in rodents, more effective than conventionally used analgesics. a** Diagram explaining the experimental timeline of **b** and **c** (created in BioRender). **b, c** Analgesic effect of LIA (10 min 365 nm illumination of the paw) in 2 chronic pain contexts: wild-type mice with a spared nerve injury (**b**), and Trek1-Trek2 KO mice (**c**). Mechanical sensitivity was quantified through von Frey withdrawal threshold. A two-sided Mann–Whitney (b−$p$ = 0.002) or two-sided Wilcoxon's signed rank test (c−$p$ = 0.0107) was used to compare pre and post-LIA von Frey withdrawal thresholds (*$p$ < 0.05, **$p$ < 0.01). **d** Diagram explaining the experimental timelines of figures (**e–i**). IP = intraperitoneal (created in BioRender). **e** Time course of the analgesic effect of LIA (10 min 365 nm illumination of the paw) on wild-type rats, quantified through von Frey withdrawal threshold. **f, g** Time course of the analgesic effect of Emla cream application (**f**) or ibuprofen injection (**g**) on wild-type mice, quantified through von Frey withdrawal threshold. For ease of comparison to LIA, von Frey withdrawal threshold evolution following LIA treatment on wild-type mice from Fig. 1f is represented in dashed magenta line. **h** Time course of the analgesic effect of LIA (10 min 365 nm illumination of the paw) on the $Trek1^{-/-}$-$Trek2^{-/-}$ KO mice chronic pain model. **i** Time course of the analgesic effects of Emla cream application and ibuprofen injection on $Trek1^{-/-}$-$Trek2^{-/-}$ KO mice, quantified through von Frey withdrawal threshold. For ease of comparison to LIA, von Frey withdrawal threshold evolution following LIA treatment on $Trek1^{-/-}$-$Trek2^{-/-}$ KO mice from (**h**) is represented in dashed magenta line. In (**e–i**), statistical analyses were carried out on each paw series independently and compare the von Frey withdrawal threshold at each time point to that at the beginning of the experiment (pre-treatment time point) using a Friedman test followed by Dunn's two-sided multiple comparison test (*$p$ < 0.05, **$p$ < 0.01 – For e: LIA-treated paws: post-LIA: $p$ = 0.0187, 1 h: $p$ = 0.0356, 3 h: $p$ = 0.4155, 6 h: $p$ = 0.3084, non-treated paws: Friedman test: $p$ = 0.0568, for (**f**): Emla-treated paws: post-Emla: $p$ = 0.1312, 1 h: $p$ = 0.7158, 3 h: $p$ > 0.9999, 6 h: $p$ = 0.8236, non-treated paws: Friedman test: $p$ = 0.5325, for (**g**): Friedman test: $p$ = 0.3389, for (**h**): LIA-treated paws: post-LIA: $p$ = 0.0107, 1 h: $p$ = 0.0048, 3 h: $p$ > 0.9999, 6 h: $p$ > 0.9999, non-treated paws: Friedman test: $p$ = 0.5861, for (**i**): Emla-treated paws: post-Emla: $p$ = 0.0046, 1 h: $p$ = 0.3587, 3 h: $p$ > 0.9999, 6 h: $p$ > 0.9999, Ibuprofen-treated paws: post-ibuprofen: $p$ = 0.0119, 1 h: $p$ = 0.264, 3 h: $p$ > 0.9999, 6 h: $p$ > 0.9999, non-treated paws: Friedman test: $p$ = 0.3696). Animal diagrams in (**e–i**) were obtained from BioRender. Data shown are mean ± s.e.m. The number of paws used in each experiment is indicated on the figure. Source data are provided in the Source Data file.

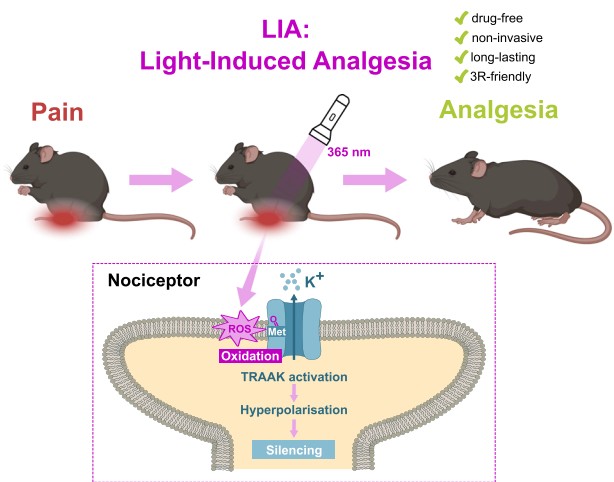

**Fig. 7 | Graphical abstract explaining LIA and its underlying mechanism.** Upon 365-nm illumination of rodent skin, TRAAK channels within the illuminated area of the skin are activated through oxidation of methionine 134. This activation hyperpolarises and silences the free nerve endings of nociceptors, thereby preventing the initiation and propagation of pain signals to the central nervous system. 365-nm skin illumination therefore results in a marked and long-lasting analgesic effect in both rats and mice.

**In vivo extracellular recordings of dorsal spinal horn neurons.** In vivo recordings were performed on Wild-Type mice. Mice were anesthetized with urethane 20% (1.5 g/kg) and placed on a stereotaxic frame (Unimécanique, Asnières, France). A laminectomy was performed on lumbar vertebrae L1–L3 and segments L4–L5 of the spinal cord were exposed. Extracellular recordings of wide dynamic range dorsal horn neurons were made with borosilicate glass capillaries (2 MΩ, filled with NaCl 684 mM) (Harvard Apparatus, Cambridge, MA, USA). The signal was amplified and high pass filtered using a DAM80 amplifier (WPI, FL, USA) connected to CED1401 (CED, UK). The acquisition was performed using spike 2 software (CED, UK). The criterion for the selection of a neuron was the presence of an A fiber-evoked response (0-80 ms) followed by a C fiber-evoked response (80 to 300 ms) to electrical stimulation of the corresponding receptive field of the ipsilateral paw with subcutaneously implanted bipolar electrodes connected to a stimulator (AMPI, Israel). The tested hind paws were exposed to 365 nm illumination during 10 min with 1-min light pulses followed by 1-min breaks to avoid any heating of the paw.

**Mechanical sensitivity measurements.** Mice and rats were placed in testing compartments set on an elevated metal mesh stand (Ugo Basile) for habituation (10 min for mice, 30 min for rats) prior to any measurements. The mechanical sensitivity of mice was then assessed using von Frey calibrated filaments (0.008 g to 8 g for mice (Ugo Basile), 4 to 26 g for rats (Bioseb))[54]. The latter were applied perpendicularly to the plantar surface of the hind paw and pressed until they bent. Filaments were used in ascending or descending order, starting with the 0.6 g filament (for the first measure of the day) or the filament that triggered a paw withdrawal in the previous von Frey experiment of the day. The von Frey withdrawal threshold was defined as the filament evoking at least 2 out of 3 (for mice) or 5 out of 5 (for rats) paw withdrawal reflexes. To test the efficiency of a given analgesic treatment, von Frey tests were carried out on the same animal, before and after the treatment.

**Spared Nerve Injury (SNI) induction (chronic pain model).** Spared nerve injury (SNI) mice were randomly assigned to SNI or sham group. Animals were anesthetized under 3% isoflurane/oxygen, the left leg sciatic nerve were exposed and the peroneal and tibial branches were ligated and transected. Skin was closed with silk sutures. Identical procedure was performed in sham operated mice except the nerve ligation. The mice were isolated until they had fully recovered from the surgery before being reintroduced with their littermate.

**365 nm illumination for LIA.** For LIA, skin 365 nm illumination was applied on the plantar surface of one of the hind paws of each mouse or rat (the contralateral side was used as a control) through the metal mesh stand, using a portable 365 nm lamp (Alonefire SV003 365 nm 10 W for mice, Alonefire SV003 365 nm 12 W for rats) set at its most powerful mode (1,39 mW/mm² for the 10 W lamp). The experimenter held the lamp 1 cm away from the paw at most and followed thoroughly the movements of the mouse. In total, LIA-treated paws were illuminated at 365 nm during 10 min with 1-min light pulses followed by 1-min breaks to avoid any heating of the paw.

**RU-TRAAK2 injection.** The RU-TRAAK2 compound was dissolved in DMSO, prior to dilution in saline solution (the extracellular solution used for patch-clamp) to prepare a 250 μM RU-TRAAK2 solution (with 1,5% DMSO). 10 μL of the latter solution was injected in the hindpaw after 365 nm illumination (LIA).

**Emla cream application.** Emla cream 5% was applied on the bottom part of the hindpaw with a cotton bud, as the mouse was held in restraint. To avoid any wiping away of the cream or any licking, mice

were maintained immobilized for 1 min at least after the end of the cream application.

**Ibuprofen injection.** Ibuprofen was administered by intraperitoneal injection of a sodium ibuprofen (I1892, Supelco) solution (17.5 mg/mL in PBS (phosphate-buffered saline)) at a dose of 60 mg/kg.

## Quantification and statistical analysis

Analyses on recorded currents were performed using Clampfit. All data were then processed and analyzed using Excel and R 4.2.0 or GraphPad Prism 8 according to the statistical test carried out. Apart from paw withdrawal threshold data, data normality was assessed with a Shapiro test. To compare 0 mV current ratios to 1, normal data were analyzed with two-sided one-sample t-tests, and non-normal data with two-sided Wilcoxon's signed rank tests. When 2 groups or conditions were compared, two-sided Wilcoxon's signed rank tests (paired data) or two-sided Mann-Whitney tests (unpaired data) were used in case of non-normal data. If data were normal and unpaired, the F-test was used to assess the homoscedasticity of the data, prior to applying two-sided unpaired Student's $t$ test (in case of equal variance) or Welch's $t$ test. Finally, in the case of normal and paired data, two-sided paired Student's $t$ test was applied. When more than 2 conditions or groups were compared, and when the data were non-normal, Kruskal–Wallis test was carried out to determine whether at least one group was significantly different than others. If it were the case, Dunn post-hoc test with Bonferroni p-value adjustment method was performed for multiple comparisons to a control group. When more than 2 conditions were compared, and in the case of normal data, Levene's test was computed to test the homoscedasticity of the data, and a one-way anova or Welch's anova was performed depending on Levene's test results. Dunnett's post-hoc test was then carried out to compare the different groups to a control group. Data on C fiber responses with respect to stimulation intensity (Fig. 5h) before or after 365 nm illumination as well as data on RU-TRAAK2 or saline injected paws (Fig. 1g) were analyzed with a 2-way Anova, followed by Holm-Sidak's multiple comparison test. As for paw withdrawal threshold data, paired non-parametric tests were automatically carried out: either two-sided Wilcoxon's ranked test when 2 conditions were compared, or Friedman's test followed by Dunn's post-hoc test for multiple comparisons to a control group.

## Reproducibility

No statistical method was used to predetermine sample size, and no data were excluded from the analyses. The experiments were not randomized, and investigators were not blinded to allocation during experiments and outcome assessment.

## Protein sequence alignment

Multiple protein sequence alignments were carried out using the Clustal Omega algorithm available on EMBL's European Bioinformatics Institute's website (https://www.ebi.ac.uk/jdispatcher/msa/clustalo) and viewed with the Jalview software.

## Methods for supplementary figure

**Histology.** Mice were euthanized 24 or 72 h after the end of LIA treatment (10 min illumination at 365 nm with a 1-min pause every minute). The skin of the hairless bottom part of both hindpaws (LIA-treated and non-treated) was carefully dissected and fixed for 4 h in Bouin's solution. Tissues were then dehydrated in an automated tissue processor (Leica TP1020) and paraffin-embedded. Sections (6 μm) obtained with a semi-automated microtome (Leica RM2245) were stained with hematoxilin-eosin (HE) and imaged in brightfield using a slide scanner (Vectra Polaris). Images were then analyzed with the Phenochart 2.0.0 software.

**Determination of the degradation timescale of the Halo-tagged mTRAAK channel pool.** To assess the degradation timescale of mTRAAK-Halo proteins, HEK cells were transiently transfected with the Halo tagged mTRAAK channel using the JetOptimus protocol (Polyplus) and split into 12-well plates. 24 h after transfection, cells were labeled with the Halo-TMR cell-permeant ligand (Promega) as described in the Promega protocol (15 min incubation in a 1 μM Halo-TMR solution in complete DMEM medium), rinsed three times with PBS to remove all unbound Halo-TMR ligands, and incubated for various times in complete DMEM medium in the dark (in the incubator 37 °C, 5% $CO_2$). Every 2.5 h for 25 h, cells from one well were harvested and lysed with RIPA lysis buffer (50 mM Tris HCl, 150 mM NaCl, 1% Triton, 0.1% sodium dodecyl sulfate (SDS), 0.5% Deoxycholate, 1% Np40, 1 mM EDTA, and phosphatase inhibitors (one PhosSTOP EASYpack tablet from Roche in 10 mL lysis buffer), pH = 8), prior to a 5-min sonication cycle. The cell lysate was then centrifugated and the supernatant was frozen at −18 °C until running a SDS-page electrophoresis of all samples.

The day of the SDS-page electrophoresis, samples were thawed on ice and the protein concentration of each sample was measured using the Bradford assay (Biorad). After chemical (in Laemmli buffer: 180 mM Tris base, 6% SDS, 30% glycerol, 0.03% bromophenol blue, 15% $\beta$-mercaptoethanol, pH = 6.8) and thermal denaturation (5 min at 95 °C), 10 μg of proteins were loaded on an SDS-page 10% acrylamide gel for electrophoresis. After migration, the fluorescence of the gel was imaged using the Amersham ImageQuant 800 imager (using a 535 nm excitation and the Cy3(UV) filter). Images were analyzed on ImageJ: the integrated fluorescence intensity of each band at ~76 kDa was subtracted to that of a portion of the corresponding lane devoid of fluorescence for background correction. Background corrected integrated intensities of each band were then divided by that of the band from the cell lysate obtained right after the labeling, yielding the mTRAAK-Halo signal ratio.

Of note, to determine the degradation timescale of mTRAAK-Halo channels in cells exposed to 365 nm light, cells were illuminated following the same illumination protocol as the one used to determine the reversibility of mTRAAK 365 nm light-induced current increase (Supplementary Fig. 3C-E), prior to Halo-TMR labelling.

**Molecular dynamics simulations. Computational model.** The initial structures of hTRAAK in its down (PDB: 4WFF) and up (PDB: 4WFE)[31] states were retrieved from the Protein Data Bank (PDB). Unresolved fragments in chain A were reconstructed by copying the corresponding resolved fragments from chain B, and vice versa. The I133M mutation was introduced to simulate the mTRAAK sequence. The proteins were protonated at pH 7.0, with their terminal ends capped using acetyl (ACE) and N-methyl amide (NME) groups. Each structure was embedded in a 1-palmitoyl-2-oleoyl-phosphatidylcholine (POPC) membrane, surrounded by water and $K^+$ and $Cl^-$ ions, to form a cubic simulation box with a salt concentration of 0.5 M. These systems were built using the CHARMM-GUI web server[55]. In the 4WFE structure, the UP conformational change is displayed only in chain B. To generate a symmetric UP state, we symmetrized the TM4 helix by copying the conformation of chain B onto chain A. Topologies and hybrid structures were generated using the *pmx*[56] Python tool with the single topology approach with dummy atoms. The coordinates of the initial and final configurations of the simulation can be found in a text file in the Source Data.

**Free energy calculations.** Simulations were done using GROMACS 2024.2[57]. Proteins were modeled with the CHARMM36m[58] force field, lipids with CHARMM36 parameters[59], water with the CHARMM TIP3P model[60], and $K^+$ and $Cl^-$ ions with Beglov and Roux parameters[61]. The parameters for L-Methionine (S)-S-oxide (referred to as Oxy-Methionine in this work) were obtained from the CHARMM36 force

field for non-standard amino acids[62]. Periodic boundary conditions were applied with a minimum solute-box distance of 1.5 nm. A thermodynamic cycle between wild-type methionine and its oxidized form was employed to calculate the relative free energy difference associated with this modification in the UP and DOWN states of hTRAAK I133M (Supplementary Fig. 7). For each state in this thermodynamic cycle, five independent replicates were performed as follows: *Energy minimization*: A steepest descent algorithm (maximum 10,000 steps) was used. *Equilibration and production*: Simulations were run in the NPT ensemble for 20 ns using the leapfrog stochastic dynamics integrator[63] with a time step of 2 fs (this integrator mitigates potential violations of energy equipartition that can arise when degrees of freedom are decoupled during rapid transitions, which may otherwise isolate them from the system and prevent kinetic energy exchange[64,65]). The pressure was kept at 1 bar with the semi-isotropic c-rescale barostat[66] and a temperature of 298.15 K using Langevin dynamics. Electrostatic interactions were computed using the particle mesh Ewald (PME) algorithm[67] with a real-space cut-off of 1.2 nm, a spline order of 4, a relative tolerance of $10^{-5}$, and a Fourier spacing of 1.2 nm[68]. The Van der Waals interactions were force switched off between 1.0 and 1.2 nm. Bond constraints were applied using the LINCS algorithm[66].

Relative free-energy differences were calculated using a non-equilibrium alchemical protocol[69,70]. Following the 20 ns of simulation after energy minimization, the first 2 ns were discarded as equilibration and the remaining 18 ns used for production. From these production runs, every fourth frame was extracted, and each selected frame served as the starting point for a 500 ps non-equilibrium trajectory in which one thermodynamic state was rapidly transformed into another. Transitions were performed in both forward and backward directions, allowing collection of the associated energetic cost (work values). In total, 113 rapid transformations were generated per production run, corresponding to 452 transformations per replicate (Supplementary Fig. 9).

During the 500 ps nonequilibrium alchemical transformations, Van der Waals and Coulomb interactions were soft-cored to prevent singularities in the potential energy surface[71] (this approach handles atomic overlapping that can happen when degrees of freedom are gradually coupled or decoupled, which would otherwise lead to large repulsive peaks in the potential (e.g., from Lennard–Jones interactions) and cause instabilities in the integration). Free energy differences were estimated using the Bennett Acceptance Ratio (BAR) method for non-equilibrium simulations[72,73], as implemented in *pmx*[56]. This implementation corresponds to a maximum-likelihood estimator based on Crooks' fluctuation theorem[73,74]. Uncertainties in free energy values were reported as standard errors of the mean across the five independent replicates, and the convergence over the simulation time was monitored as well (Supplementary Fig. 10).

## Reporting summary
Further information on research design is available in the Nature Portfolio Reporting Summary linked to this article.

## Data availability
Source data are provided with this paper and are also available on the figshare repository: https://doi.org/10.6084/m9.figshare.30788849.

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

## Acknowledgements

The authors thank Brigitte Wdziekonski, Vanessa Roubeaud for their technical help, as well as Stéphane Supplisson, Dominique Debanne and Hélène Guizouarn for their insightful discussions and comments. This work was supported by grants to Guillaume Sandoz by the Fondation Simone et Cino del Ducas (Institut de France), Agence Nationale pour la Recherche (Rhodopsins, ANR-19-CE11-0026-01; Electro-Wnt, ANR-24-CE14-7899; HumASIC, ANR-22-CE16-0006), the Laboratory of Excellence "Ion Channel Science and Therapeutics" (grant ANR-11-LABX-0015-01), by the French government, through the France 2030 investment plan managed by the Agence Nationale de la Recherche, as part of the Université Côte d'Azur's Initiative of Excellence (reference ANR-15-IDEX-01), by a grant to Johannes Broichhagen by European Union's Horizon Europe Framework Programme (deuterON, grant agreement no. 101042046), by grants to Eric Boué-Grabot by Agence Nationale pour la Recherche (PurplePAIN, ANR-20-CE14-0016) and "Investments for the Future" programs of the University of Bordeaux's IdEx program GPR BRAIN_2030, by a grant to Felix Viana by Ministerio de Ciencia e Innovación (GVA-PROMETEO/2021/03, PID2022-140961OB-I00, CEX2021-001165-S), and by a grant to Marion Bied by Fondation pour la recherche médicale (FDT202404018660).

## Author contributions

Conceptualization, M.B., A.L.W., and G.S.; Methodology, M.B., A.L.W., S.J.T., J.N., W.K., F.V., J.B., E.B.G., and G.S.; Investigation, M.B., A.L.W., A.A.C., E.F.M.O., B.S., K.R., E.P.G., P.F., and W.K.; Writing—original draft, M.B. and G.S.; Writing—review & editing, M.B., A.L.W., E.P.G., S.J.T., J.N., F.V., J.B., E.B.G., and G.S.; Funding acquisition, G.S.; Project administration, G.S.; Supervision, G.S.

## Competing interests

The authors declare no competing interests.
