## [Transparent Peer Review file · Nature Communications]

Light-Induced Analgesia Provides A Drug-Free Optical Method for Pain Relief via Activation of TRAAK K⁺ Channels

Corresponding Author: Dr Guillaume Sandoz

Version 0:

Reviewer comments:

Reviewer #1

(Remarks to the Author)

The manuscript describes a comprehensive study focusing on using an innovative UV-A light-induced analgesia approach for mice and rats, which originates from the activation of the leak ion channel TRAAK from a two-pore domain potassium channel (K2P) family. The activation of this ion channel leads to hyperpolarization of the neuronal membrane potential resulting in an inhibition of nociceptive signal propagation. The authors were able to trace TRAAK activation to an oxidation of a specific methionine residue at the TRAAK regulatory fenestration site upon UV-A light illumination. This effect is species-dependent, with many rodents but not primates are prone to such activation, which the authors traced to the regulatory methionine substitution to other residues in UV-A insensitive TRAAK channels (e.g., human TRAAK). They demonstrated that the effect can be also mimicked even without UV-A exposure by methionine mutation to electron-rich glutamate or glutamine residues. The UV-A sensitivity can be also transferred to other K2P family channels by relevant regulatory residue substitution to methionine. The authors established in the *in vivo* studies that UV-A light-induced analgesia approach can be even more effective than more traditional, complicated and/or invasive approaches using non-steroidal anti-inflammatory (NSAID) ibuprofen injections or special analgesic cream applications.

The manuscript is written very well, very concisely and packs a large number of diverse experimental and computational approaches - from patch-clamp electrophysiology, *in vivo* animal studies, histology to multiple protein sequence alignments and molecular dynamics simulations. All experiments are described very well. The figures are also of very high quality and represent the results of the study very well. The description of experimental results is very accessible for general readership. The study is also of a high significance for biomedical research. In my opinion, this manuscript is neatly ready for publication.

My relatively minor concern is that I do not quite understand why such a seemingly non-selective method such as UV-A light exposure leads to a very particular result, which is oxidation of a specific methionine residue in a specific TRAAK ion channel protein. I would expect that the effect of such exposure would be much more widespread and would involve multiple residue oxidation, not only methionine but also cysteine and possibly tryptophan and tyrosine as well. UV-A light can also cause lipid peroxidation and even some DNA damage (although obviously not as much as UV-B or UV-C). In any case, the effect on a very specific residue in one ion channel protein is very surprising although the authors were able to demonstrate such selectivity through multiple experiments. I wonder since the methionine residue is located near a lipid facing fenestration, a "perfect storm" of both residue and lipid oxidation can cause a cumulative effect. That would be interesting to test, although might be out of scope of this particular study. One optional suggestion for a future work would be to test the simultaneous effect of the Met mutation to Glu and Gln with added UV-A exposure, if this would lead even to a more profound effect. One can also potentially test the Met mutation to a positively charged residue, e.g. Arg, which would likely inhibit TRAAK activation based on the authors' arguments.

Another minor suggestion is related to molecular dynamics (MD) simulation description. It may need to be improved by adding references for force fields and other methodology used, pmx program (which I guess is this one <https://bioexcel.eu/pmx-new/>) etc. I was also confused by sentences like "leapfrog stochastic dynamics integrator". It does not seem that stochastic dynamics was used in this work. Rather all-atom enhanced sampling MD simulations were employed using free energy perturbation (FEP) methodology. There is some confusion about simulation times etc. Also, only one number for a relative free energy change is shown in the Extended Data Fig. 6: It would be helpful to see individual free energy changes for "up" and "down" systems and how they converge over simulation times, for instance. It would be helpful

to know if a single (with dummy atoms) or dual topology approach was used for FEP simulations.

There are a couple of other minor text and figure related issues:

p 11 "Whereas optogenetic and photopharmacology use genetic or pharmacology," - this is confusing

Extended data Fig. 5A. Chemical formulas shown are for isolated amino acids, not for amino acid residues, which may cause some confusion.

Reviewer #2

(Remarks to the Author)

The manuscript "Light-Induced Analgesia: A Drug Free Optical Method for Pain Relief via Activation of TRAAK K⁺ Channels" presents a novel, drug-free, and non-invasive method for pain relief in rodents using ultraviolet-A (UV-A, 365 nm) light to activate the TRAAK potassium channel. The mechanism is based on the oxidation of a specific methionine residue (M134) in the TRAAK channel, which triggers its activation and silences nociceptor activity, resulting in potent local analgesia. The authors demonstrate that this effect is robust in mice and rats, outperforms standard analgesics, and aligns with the ethical principles of animal research. This study represents a significant conceptual advancement in the field of pain research. By proposing an optical analgesia technique devoid of pharmacological agents, it offers a valuable alternative for preclinical models, potentially reducing off-target effects and pharmacodynamic variability. The authors meticulously dissect the molecular underpinnings of TRAAK activation, showing that oxidation of M134 is both necessary and sufficient for light sensitivity. They further employ a multifaceted experimental approach, integrating electrophysiology, behavioral assays, site-directed mutagenesis, and computational modeling to substantiate their findings. Importantly, interspecies comparisons of TRAAK channel behavior provide insight into evolutionary divergence and underscore the selective light sensitivity observed in rodent models.

Concerns:

1. Species Specificity: The UV-A sensitivity of TRAAK is not present in humans or several other mammals; it is limited to rodents and a few other species. This restricts the translational value of the method for human pain therapy.
2. Requirement for Genetic Modification: For human TRAAK to gain UV-A sensitivity, a specific mutation (I133M, homologous to mouse M134) must be introduced. This would require genetic modification or gene therapy, which is currently not practical or ethical for routine pain management in humans.
3. Potential Safety Concerns: While the illumination protocol is shown to be safe in rodents, the long-term effects of repeated UV-A exposure, are not addressed. UV-A can cause DNA damage and skin aging.
4. Temperature effects. The experiments demonstrating TRAAK activation in heterologous expression cells were all performed at room temperature. It is not clear whether the degree of activation by UV-A would become significantly attenuated when the experiments were performed at body temperatures of 32-37°C.
5. Lack of Information on Nociceptors: All recordings of TRAAK-mediated currents and effects of UV-A were performed in heterologous expression cells. The authors did not investigate whether leak currents mediated by TRAAK were enhanced and neuronal excitability decreased in nociceptors.

Version 1:

Reviewer comments:

Reviewer #1

(Remarks to the Author)

The manuscript presents a thorough investigation of a novel UV-A light-induced analgesia (LIA) approach in mice and rats, based on the activation of the leak potassium channel TRAAK, a member of the K2P two-pore-domain potassium channel family. UV-A illumination triggers TRAAK activation, which hyperpolarizes neuronal membranes and thus suppresses nociceptive signal transmission. The authors identify this UV-A sensitivity as arising from oxidation of a specific methionine residue M134 on the transmembrane helix 2 (TM2) facing crucial regulatory fenestration site within TRAAK. Oxidation of this methionine by UV-A induced reactive oxygen species (ROS) leads to TRAAK activation. This was confirmed by alchemical free energy perturbation calculations based on molecular dynamics (MD) simulations. The authors discovered that this regulation mechanism is species-dependent. Rodent TRAAK channels contain this methionine and are UV-A responsive, whereas primate TRAAK channels, including human TRAAK, are not. Introducing methionine into UV-A-insensitive TRAAK channels confers UV-A sensitivity, while substituting methionine with polar and charged residues such as arginine, glutamate or glutamine can mimic channel activation even in the absence of UV-A light. Furthermore, transferring the corresponding methionine residue to other K2P channels imparts similar UV-A responsiveness. In vivo studies demonstrate that this UV-A-based analgesic approach can surpass in effectiveness traditional and more invasive treatments, such as ibuprofen injections or topical analgesic applications.

The manuscript is very well written and represents an important contribution to biomedical research, in particular, animal model studies. While a direct translation of this study to human health might not be feasible, the presented approach and findings can be potentially extended to other similar techniques such as light stimulation therapy and identification of novel protein targets for pain therapy. This is the revised version of the manuscript, and the authors provided very thorough

responses to the identified shortcomings, performed several new experiments, significantly extended description of computer simulation methodology and added several supporting figures as well. I do not have additional comments and would recommend acceptance of the manuscript in its current form.

Reviewer #2

(Remarks to the Author)

The authors have very well responded my comments. I have no further concerns, except that the animal species should be specified in the title of the manuscript.

We thank the reviewers for their helpful remarks and suggestions. In response, we have carried out additional experiments, that we integrated in the manuscript and clarified the text according to the reviewers' comments. Altogether, we believe that the additional experiments and modifications to the text bolster our conclusions and considerably strengthen the manuscript.

Herewith we submit for your consideration a point-by-point response to each of the reviewers' concerns and we explain the changes that we have made as required.

Reviewer #1 (Remarks to the Author)

The manuscript describes a comprehensive study focusing on using an innovative UV-A light-induced analgesia approach for mice and rats, which originates from the activation of the leak ion channel TRAAK from a two-pore domain potassium channel (K2P) family. The activation of this ion channel leads to hyperpolarization of the neuronal membrane potential resulting in an inhibition of nociceptive signal propagation. The authors were able to trace TRAAK activation to an oxidation of a specific methionine residue at the TRAAK regulatory fenestration site upon UV-A light illumination. This effect is species-dependent, with many rodents but not primates are prone to such activation, which the authors traced to the regulatory methionine substitution to other residues in UV-A insensitive TRAAK channels (e.g., human TRAAK). They demonstrated that the effect can be also mimicked even without UV-A exposure by methionine mutation to electron-rich glutamate or glutamine residues. The UV-A sensitivity can be also transferred to other K2P family channels by relevant regulatory residue substitution to methionine. The authors established in the in vivo studies that UV-A light-induced analgesia approach can be even more effective than more traditional, complicated and/or invasive approaches using non-steroidal anti-inflammatory (NSAID) ibuprofen injections or special analgesic cream applications.

The manuscript is written very well, very concisely and packs a large number of diverse experimental and computational approaches - from patch-clamp electrophysiology, in vivo animal studies, histology to multiple protein sequence alignments and molecular dynamics simulations. All experiments are described very well. The figures are also of very high quality and represent the results of the study very well. The description of experimental results is very accessible for general readership. The study is also of a high significance for biomedical research. In my opinion, this manuscript is neatly ready for publication.

We sincerely thank the reviewer for his/her very positive and encouraging assessment of our work.

My relatively minor concern is that I do not quite understand why such a seemingly non-selective method such as UV-A light exposure leads to a very particular result, which is oxidation of a specific methionine residue in a specific TRAAK ion channel protein. I would expect that the effect of such exposure would be much more widespread and would involve multiple residue oxidation, not only methionine but also cysteine and possibly tryptophan and tyrosine as well. UV-A light can also cause lipid peroxidation and even some DNA damage (although obviously not as much as UV-B or UV-C). In any case, the effect on a very specific residue in one ion channel protein is very surprising although the authors were able to demonstrate such selectivity through multiple experiments.

We agree with the reviewer that it is indeed surprising that the entire effect depends on the oxidation of a single residue, methionine M134 in mTRAAK. This observation provides additional evidence for the functional importance of the regulatory fenestration site harboring M134. In our case, activation is triggered by the oxidation of this native methionine, which induces a conformational switch from the inactive (down) to the active (up) state. While unexpected, our finding highlights the pivotal regulatory role of M134 in channel activity. This conclusion is further supported by our gain-of-function mutants (mTRAAK M134E and mTRAAK M134D), obtained by mutating this single residue. In line with this, previous reports have also demonstrated that single-point mutations within the fenestration site can strongly modulate channel activity ((I) Bauer et al (2018). Mutations in KCNK4 that Affect Gating Cause a Recognizable Neurodevelopmental Syndrome. *The American Journal of Human Genetics* **103**, 621–630. <https://doi.org/10.1016/j.ajhg.2018.09.001>, (II) Ben Soussia et al (2019). Mutation of a single residue promotes gating of vertebrate and invertebrate two-pore domain potassium channels. *Nat Commun* **10**, 787, <https://doi.org/10.1038/s41467-019-08710-3>).

I wonder since the methionine residue is located near a lipid facing fenestration, a "perfect storm" of both residue and lipid oxidation can cause a cumulative effect. That would be interesting to test, although might be out of scope of this particular study. One optional suggestion for a future work would be to test the simultaneous effect of the Met mutation to Glu and Gln with added UV-A exposure, if this would lead even to a more profound effect. One can also potentially test the Met mutation to a positively charged residue, e.g. Arg, which would likely inhibit TRAAK activation based on the authors' arguments.

We thank the reviewer for this interesting hypothesis which we have tested. As shown on **Fig. R1 b, c**, UV-A exposure (365 nm for 100 s) does not have any additional activating effect on either mTRAAK M134E nor mTRAAK M134Q, thus excluding the possibility of any cumulative effect of lipid oxidation that would affect mTRAAK at its fenestration site. Please note that these results are in line with the UV-A insensitivity of mTRAAK M134I (**Fig. 3 e-g**).

Figure R1: mTRAAK M134 mutants are not activated by 365 nm light.

a. Comparison of the basal 0 mV currents of mTRAAK, mTRAAK M134E, mTRAAK M134Q and mTRAAK M134R. Statistical analyses compare the 0 mV currents of mTRAAK mutants to that of wild type mTRAAK using a Kruskal-Wallis test followed by Dunn's multiple comparison test (** $p < 0.001$, **** $p < 0.0001$).

b. Time course of responses of mTRAAK M134 mutant channels to a 100 s 365 nm illumination. Current fold increases were computed at 0 mV relative to the pre-illumination 0 mV current of the cell.

c. Current fold increases at 0 mV for the mTRAAK M134 mutants after 100 s of 365 nm-centred illumination. Two independent statistical analyses were carried out: one-sample t-tests or one-sample Wilcoxon tests comparing fold increases to 1 (## $p < 0.01$, ##### $p < 0.0001$) and Kruskal-Wallis tests followed by Dunn's multiple comparison test comparing responses to that of mTRAAK (**** $p < 0.0001$).

As suggested by Rev1, we also studied the mTRAAK M134R mutant in which M134 was replaced by a positively charged residue (**Fig. R1**). Similar to mTRAAK M134E or M134Q mutants, mTRAAK M134R features a significantly higher basal current compared to wild type mTRAAK (more than 20-fold higher), but is lower than that for the M134E and M134Q mutants. As expected, mTRAAK M134R is also not activated by 365 nm illumination (**Fig. R1 b, c**). These data therefore confirm that either a polar or charged residue at this specific fenestration site results in promotion of the activated state of the channel, no matter the nature of the charge (positive or negative).

We have now added the results on mTRAAK M134R in **Fig. 4e** and **Extended Fig. 6** and added the sentence as follows:

“Similar to mTRAAK M134E or M134Q mutants, we found that introducing a positively charged residue (M134R) activates mTRAAK but not as much as a partially or full negatively charged residue (**Fig. 4e, Fig. 6b,c**). Our results thus confirm that introduction of a charged or polar environment at the position of M134 is associated with an activated state of mTRAAK, no matter the nature of the charge.”

New Fig. 4: Light activation of TRAAK relies on the oxidation of methionine 134 which switches TRAAK to its conductive “up” conformation

a. Diagram highlighting the properties of the CA-ATTO-MB2 compound, enabling the spatio-temporal control of singlet oxygen generation.

b. Representative normalized whole-cell current traces of HEK293T cells expressing Halo-tagged mTRAAK or mTRAAK M134I before and after a 25 s 565 nm illumination, as cells were previously incubated or not during 30 min in 1 μ M CA-ATTO-MB2. Currents were elicited by voltage-clamp ramps (-100 to +100 mV, 800 ms) and were normalized by the 0 mV current of the cell recorded before illumination.

c. Time course of responses of HEK293T cells expressing Halo-tagged mTRAAK or mTRAAK M134I to a 25 s 565 nm illumination, as the cells were previously incubated or not during 30 min in 1 μ M CA-ATTO-MB2.

d. Current fold increases at 0 mV of Halo-tagged mTRAAK or mTRAAK M134I expressing HEK cells previously incubated or not in 1 μ M CA-ATTO-MB2, after a 25 s 565 nm illumination. Two independent statistical analyses were carried out: two-sided one sample t-tests comparing 0 mV current fold increases to 1 (# $p < 0.05$, ## $p < 0.01$, ### $p < 0.001$, #### $p < 0.0001$) and Kruskal-Wallis test followed by Dunn's multiple comparison test comparing fold increases to that of Halo-mTRAAK expressing cells incubated in 1 μ M CA-ATTO-MB2 (* $p < 0.05$, ** $p < 0.01$, *** $p < 0.001$, **** $p < 0.0001$).

e. Comparison of the basal 0 mV currents of mTRAAK, mTRAAK M134E, mTRAAK M134Q and mTRAAK M134R. Statistical analyses compare the 0 mV currents of mTRAAK mutants to that of wild type mTRAAK using a Kruskal-Wallis test followed by Dunn's multiple comparison test (**** $p < 0.0001$).

f. Structural models of hTRAAK I133M with M133 in its reduced (left) or oxidised form (right) impacting the ("down" or "up") conformational state of the channel. Insets show the ribbon structure of hTRAAK I133M, with the black outline indicating the zoomed-in region of the TM2-TM4 interface. As highlighted in the figure, this interface undergoes substantial changes upon the down to up conformational switch of the channel. The transmembrane helix TM2(A) is highlighted in cyan, and TM4(B) is shown in dark gray. In the down state (left), the fenestration site is indicated with an arrow.

New Extended Data Fig. 6: **Oxidation of methionine 134 modifies the electronic density at its core, promoting the activated-state of TRAAK.**

- a. Oxidation of methionine into methionine sulfoxide. Skeletal formulas of methionine and methionine sulfoxide are represented, highlighting the addition of a double-bond oxygen in methionine sulfoxide which results in a modified electronic density. (ROS : Reactive Oxygen Species).
- b. Skeletal formulas of glutamate, glutamine and arginine, three polar amino acids, featuring a negative full (glutamate) or partial charge (glutamine), or a positive charge (arginine).
- c. Representative whole-cell current traces of HEK293T cells expressing mTRAAK, mTRAAK M134E, mTRAAK M134Q or mTRAAK M134R. Currents were elicited by voltage-clamp ramps (-100 to +100 mV, 800 ms).

Another minor suggestion is related to molecular dynamics (MD) simulation description. It may need to be improved by adding references for force fields and other methodology used, pmx program (which I guess is this one <https://bioexcel.eu/pmx-new/>) etc.

We thank the reviewer for this comment. We have now included the appropriate citations for the force fields and methodologies employed, as well as the reference describing the implementation of *pmx*.

I was also confused by sentences like "leapfrog stochastic dynamics integrator".

We thank the reviewer for spotting this. We recognize that the lack of detail in our original description may have caused confusion by inclusion of this term without appropriate reference. By “leapfrog stochastic dynamics integrator”, we mean that during our transitions (when interpolating between Hamiltonians) we do not employ the standard leapfrog integrator to propagate Newton’s equations of motion. Instead, we use an integrator that incorporates both friction and noise terms (stochastic dynamics).

The rationale is as follows: when degrees of freedom are coupled or decoupled in the simulation (e.g., by turning non-bonded interactions on or off to interpolate between Hamiltonians), these degrees of freedom may become isolated from the rest of the system, hindering kinetic energy exchange and potentially violating equipartition. To prevent this artifact, each degree of freedom is independently coupled to a thermal bath, which is ensured by the stochastic dynamics integrator. Specifically, we use the 'sd' integrator in the gromacs parameter file, as described in detail here: <https://manual.gromacs.org/documentation/current/reference-manual/algorithms/stochastic-dynamics.html>.

To avoid confusion, we have now clarified this point in the methodology and we have included the relevant references.

It does not seem that stochastic dynamics was used in this work. Rather all-atom enhanced sampling MD simulations were employed using free energy perturbation (FEP) methodology.

We thank the reviewer for this insightful comment and agree that indeed we used an enhanced sampling MD simulation strategy. Specifically, we calculated relative free energy differences between two thermodynamically defined states, A and B, in a manner related to, though distinct from, the classical free energy perturbation (FEP) methodology. We recognize that our initial description of this distinction may have been unclear and could have caused confusion.

In contrast to FEP, which relies on equilibrium sampling at fixed values of the λ coordinate interpolating between wild-type (WT) and mutated (Oxy) Hamiltonians, our protocol is based on rapid non-equilibrium transitions. Here, λ changes continuously over a finite transition time from 0 to 1 (i.e., from WT to Oxy Hamiltonians). During each transition, we record the derivative of the Hamiltonian with respect to λ and integrate this quantity to obtain the dissipated work. Finally, by applying an estimator based on the Crooks' fluctuation theorem, we can estimate the free energy difference from the intersection of the work distributions obtained from forward (WT \rightarrow Oxy) and reverse (Oxy \rightarrow WT) transitions in both the UP and DOWN states. Under equilibrium conditions, this approach reduces to the classical FEP framework.

We would like to emphasize that this alchemical non-equilibrium free energy calculation methodology, as well as the use of stochastic dynamics, have been previously thoroughly discussed and tested (see: (I) Gapsys et al (2016). Accurate and Rigorous Prediction of the Changes in Protein Free Energies in a Large-Scale Mutation Scan. *Angew Chem Int Ed* **55**, 7364–7368. <https://doi.org/10.1002/anie.201510054>, (II) Gapsys, et al (2020). Large scale relative protein ligand binding affinities using non-equilibrium alchemy. *Chem. Sci.* **11**, 1140–1152. <https://doi.org/10.1039/c9sc03754c>. (III) Gapsys, et al (2014). Calculation of Binding Free Energies. *Methods in Molecular Biology*, 173–209. https://doi.org/10.1007/978-1-4939-1465-4_9. (IV) Yee, et al. (2019). A molecular mechanism for transthyretin amyloidogenesis. *Nat Commun* **10**. <https://doi.org/10.1038/s41467-019-08609-z>).

We have therefore followed established best practice (see e.g. (I) Hahn, et al. (2022). Best Practices for Constructing, Preparing, and Evaluating Protein-Ligand Binding Affinity Benchmarks [Article v1.0]. *LiveCoMS* **4**. <https://doi.org/10.33011/livecoms.4.1.1497>)

Thus to answer the reviewers concerns, we have now expanded the explanation of the methodology in the main manuscript and included citations to the relevant articles.

There is some confusion about simulation times etc. Also, only one number for a relative free energy change is shown in the Extended Data Fig. 6: It would be helpful to see individual free energy changes for "up" and "down" systems and how they converge over simulation times, for instance.

We thank the reviewer once again for their helpful comments. We have now added an extra diagram in the Extended Data Figures that outlines the protocol followed (**Extended Data Fig. 8**), including the corresponding simulation times. In addition, we have included a Figure (**Extended Data Fig. 9**) showing the convergence of our free-energy estimates for the up and down states, together with their respective relative differences.

Extended Data Fig. 8: Non-equilibrium free energy calculation protocol.

Protocol for estimating the relative free-energy change of the M133 (WT) to OxyM133 (methionine sulfoxide) transformation. Each 500 ps fast transition converts methionine into its oxidized form and vice versa. The value of the dissipated work is obtained by integrating the energy change over the transformed λ coordinate. Multiple forward and backward transitions yield work distributions, whose intersection provides the free-energy estimate via Crooks' fluctuation theorem. The procedure was performed in both UP and DOWN hTRAAK states, with the I133M mutation, across five independent replicates.

Extended Data Fig. 9: Convergence of free-energy estimates over the sampling time.

(Left) the relative difference between the free-energy change of methionine oxidation in the DOWN state and that in the UP state; (center) the free-energy change of methionine oxidation in the DOWN state; (right) the free-energy change of methionine oxidation in the UP state. Error bars represent the standard error of the mean across five independent replicates. Eq: equilibrium.

It would be helpful to know if a single (with dummy atoms) or dual topology approach was used for FEP simulations.

We have now clarified in the methodology section that a single-topology approach was employed with dummy atoms for the alchemical non-equilibrium free energy estimation.

There are a couple of other minor text and figure related issues:

p 11 "Whereas optogenetic and photopharmacology use genetic or pharmacology," - this is confusing

We have replaced this portion of sentence in the text with: "Contrary to current light-control techniques that depend on genetics (optogenetics) or pharmacology (photopharmacology),"

Extended data Fig. 5A. Chemical formulas shown are for isolated amino acids, not for amino acid residues, which may cause some confusion.

We thank the reviewer for pointing out this confusion and have modified the figure accordingly.

Reviewer #2 (Remarks to the Author)

The manuscript "Light-Induced Analgesia: A Drug Free Optical Method for Pain Relief via Activation of TRAAK K⁺ Channels" presents a novel, drug-free, and non-invasive method for pain relief in rodents using ultraviolet-A (UV-A, 365 nm) light to activate the TRAAK potassium channel. The mechanism is based on the oxidation of a specific methionine residue (M134) in the TRAAK channel, which triggers its activation and silences nociceptor activity, resulting in potent local analgesia. The authors demonstrate that this effect is robust in mice and rats, outperforms standard analgesics, and aligns with the ethical principles of animal research. This study represents a significant conceptual advancement in the field of pain research. By proposing an optical analgesia technique devoid of pharmacological agents, it offers a valuable alternative for preclinical models, potentially reducing off-target effects and pharmacodynamic variability. The authors meticulously dissect the molecular underpinnings of TRAAK activation, showing that oxidation of M134 is both necessary and sufficient for light sensitivity. They further employ a multifaceted experimental approach, integrating electrophysiology, behavioral assays, site-directed mutagenesis, and computational modeling to substantiate their findings. Importantly, interspecies comparisons of TRAAK channel behavior provide insight into evolutionary divergence and underscore the selective light sensitivity observed in rodent models.

We sincerely thank the reviewer for his/her very positive and encouraging assessment of our work.

Concerns:

1. *Species Specificity: The UV-A sensitivity of TRAAK is not present in humans or several other mammals; it is limited to rodents and a few other species. This restricts the translational value of the method for human pain therapy.*

We agree with the reviewer that the insensitivity of hTRAAK to UV-A light obviously restricts the translational value of illuminating the human skin with 365 nm light to induce analgesia. Nevertheless, our work does demonstrate that specifically activating TRAAK results in analgesia, a finding that should also be relevant to TRAAK in humans even if it is UV-insensitive. Therefore, we demonstrate that TRAAK represents a potential target of choice for the development of a new class of analgesics, constituting a translational value of our study for human pain therapy. The translational value of this approach to rodents should also not be underestimated given the many millions of ear clips and tail biopsies that are taken around the world every year for genotyping purposes alone.

2. *Requirement for Genetic Modification: For human TRAAK to gain UV-A sensitivity, a specific mutation (I133M, homologous to mouse M134) must be introduced. This would require genetic modification or gene therapy, which is currently not practical or ethical for routine pain management in humans.*

We agree that the UV-A-based method developed here to induce analgesia would be more difficult to apply to humans due to the lack of a methionine residue at the appropriate site and so would require genetic modification. We have attempted to be careful in our conclusions, as this specific approach is clearly not intended for pain relief in humans. However, similar to our response to the previous question it clearly identifies activation of TRAAK as a potential therapeutic strategy in humans as well as identifying a potent and drug-free analgesic strategy in rodents that is well suited for preclinical research and veterinary applications.

3. *Potential Safety Concerns: While the illumination protocol is shown to be safe in rodents, the long-term effects of repeated UV-A exposure, are not addressed. UV-A can cause DNA damage and skin aging.*

The reviewer raises an important point about UV-A light and DNA damage etc. However, the wavelength we use here (365 nm) is similar to the UV lights used in some night clubs which is shifted to the lower energy limit of UV-A light, thus limiting its damaging effects. Moreover, in the context of animal experimentation, the animals are only exposed for a relatively short period of time that would not be expected to cause skin damage and no inflammation is seen as shown in **Extended Fig2**.

To clarify this, we have added the following sentence in the text:

“Potential long-term side-effects (premature skin aging, DNA damage) due to the repeated application of this illumination protocol cannot however be fully excluded. Nevertheless, they

are less of an issue in the scope of preclinical studies given the relatively short lifetime of rodents used in research settings.”

4. *Temperature effects. The experiments demonstrating TRAAK activation in heterologous expression cells were all performed at room temperature. It is not clear whether the degree of activation by UV-A would become significantly attenuated when the experiments were performed at body temperatures of 32-37°C.*

We thank the reviewer for this important point. In line with this comment, we carried out our standard 365 nm illumination protocol at 32°C (corresponding to skin temperature (Mei et al (2018). Body temperature measurement in mice during acute illness: implantable temperature transponder versus surface infrared thermometry. *Sci Rep* **8**, 3526)) on mTRAAK expressed in HEK cells. Similar to cells recorded at room temperature (24°C), mTRAAK currents at 32°C also markedly increased upon 365 nm light illumination.

We now have added a new **Extended Data Fig. 1** and modified the text in the manuscript as follows:

“Intriguingly, we observed that whereas no or moderate current increases were observed for TREK1, TREK2, TASK1, TASK3 and TRESK, cells expressing TRAAK showed a robust ~8-fold current increase at 0 mV after 100 s of 365 nm illumination at 0.47 mW/mm² (**Fig. 1a-c**), at both room temperature (24°C) and skin temperature (32°C) ³ (**Extended Data Fig. 1**).”

Extended Data Fig. 1: 365 nm light activation of mTRAAK occurs similarly at room (24°C) and skin (32°C) temperature.

a. Representative whole-cell current traces of HEK293T cells expressing mTRAAK recorded at 24°C or 32°C, before and after a 100 s 365 nm illumination. (Traces for one given temperature were obtained from the same cell.)

b. Time course of mTRAAK responses to a 100 s 365 nm-centred illumination recorded at 24°C or 32°C. Current fold increases were computed at 0 mV relative to the pre-illumination 0 mV current of the cell.

c. Comparison of mTRAAK 0 mV current densities at 24°C or 32°C, before and after a 100 s 365 nm-centred illumination (carried out at 24°C or 32°C accordingly). Statistical analyses compare the 0 mV current densities of mTRAAK before and after the illumination either at 24°C or 32°C using a Bonferroni-corrected paired t-test (*** $p < 0.0001$).

5. *Lack of Information on Nociceptors: All recordings of TRAAK-mediated currents and effects of UV-A were performed in heterologous expression cells. The authors did not investigate whether leak currents mediated by TRAAK were enhanced and neuronal excitability decreased in nociceptors.*

We thank the reviewer for this valuable comment and for pointing out the suggested experiment. We agree that directly demonstrating the effect of 365 nm light on neuronal excitability and leak currents in nociceptors would provide even stronger support for our findings on LIA. However, direct *in vivo* patch-clamp recordings from DRG neurons remain technically challenging. These neurons are highly specialized, with a complex structural and molecular organization, and *in vivo* it is the terminal nerve endings that are exposed to light and express channels with sensory transduction roles. At these endings, even a local increase in K^+ conductance can strongly counteract the activation of depolarizing cationic channels such as Transient Receptor Potential channels (TRPs).

While patch-clamp experiments on dissociated DRG cultures are perhaps feasible, they lack the anatomical organization of peripheral extensions, making it impossible to directly assess channel function at the level of nerve terminals. For this reason, we focused instead on nerve–skin preparations, which preserve the native structure of nociceptors and therefore a better physiological readout. As shown in **Fig. 5 a-c**, 365 nm illumination directly altered nociceptor activity by significantly increasing the threshold for electrical stimulation. In addition, our *in vivo* experiments (**Fig. 5 d-h**) provided an indirect but physiologically relevant readout of nociceptor excitability, and again demonstrated a reduction in neuronal responsiveness following UV-A exposure. Furthermore, the use of a specific TRAAK inhibitor *in vivo* to

abrogate the analgesic response to UV-A exposure (**Fig 1d, g**) demonstrates a direct role for TRAAK.